# Therapeutic Potential of Exosomes in Tendon and Tendon–Bone Healing: A Systematic Review of Preclinical Studies

**DOI:** 10.3390/jfb14060299

**Published:** 2023-05-28

**Authors:** Mingrui Zou, Jingzhou Wang, Zhenxing Shao

**Affiliations:** 1Department of Sports Medicine, Peking University Third Hospital, Institute of Sports Medicine of Peking University, Beijing 100191, China; 2110301145@stu.pku.edu.cn (M.Z.); 2010301228@stu.pku.edu.cn (J.W.); 2Beijing Key Laboratory of Sports Injuries, Engineering Research Center of Sports Trauma Treatment Technology and Devices, Ministry of Education, Beijing 100191, China

**Keywords:** exosomes, extracellular vesicles, regenerative medicine, tendon healing, tendon–bone healing

## Abstract

Exosomes have been proven to play a positive role in tendon and tendon–bone healing. Here, we systematically review the literature to evaluate the efficacy of exosomes in tendon and tendon–bone healing. Following the Preferred Reporting Items for Systematic Reviews and Meta-Analyses guidelines, a systematic and comprehensive review of the literature was performed on 21 January 2023. The electronic databases searched included Medline (through PubMed), Web of Science, Embase, Scopus, Cochrane Library and Ovid. In the end, a total of 1794 articles were systematically reviewed. Furthermore, a “snowball” search was also carried out. Finally, forty-six studies were included for analysis, with the total sample size being 1481 rats, 416 mice, 330 rabbits, 48 dogs, and 12 sheep. In these studies, exosomes promoted tendon and tendon–bone healing and displayed improved histological, biomechanical and morphological outcomes. Some studies also suggested the mechanism of exosomes in promoting tendon and tendon–bone healing, mainly through the following aspects: (1) suppressing inflammatory response and regulating macrophage polarization; (2) regulating gene expression, reshaping cell microenvironment and reconstructing extracellular matrix; (3) promoting angiogenesis. The risk of bias in the included studies was low on the whole. This systematic review provides evidence of the positive effect of exosomes on tendon and tendon–bone healing in preclinical studies. The unclear-to-low risk of bias highlights the significance of standardization of outcome reporting. It should be noted that the most suitable source, isolation methods, concentration and administration frequency of exosomes are still unknown. Additionally, few studies have used large animals as subjects. Further studies may be required on comparing the safety and efficacy of different treatment parameters in large animal models, which would be conducive to the design of clinical trials.

## 1. Introduction

Tendons are structures composed of fibrous connective tissue that transmit power from muscles to bones. Although tendons can withstand different loadings, their damage is extremely severe. Due to the lack of blood vessels, the regeneration ability of tendons is poor. Therefore, tendon repair is a huge challenge in medicine. Tendon or ligament injuries are common. In the United States, every year there are more than 300,000 patients who require surgery to repair tendons or ligaments, and the number is growing as the population grows and sports are becoming more popular [1,2].

Common tendon injury can be divided into chronic injury caused by degeneration and acute injury caused by direct rupture, both of which significantly change the structure and function of the tendon. In severe cases, the tendon–bone interface is also damaged [3]. Generally, four stages take place in tendon or tendon–bone healing. The first stage is the inflammation stage, when macrophages are recruited to the injured site and vascular permeability increases. The second stage is the proliferation stage, which takes a long time. In this stage, cells proliferate and produce some growth factors. The next stage is the remodeling stage, when fibrocartilage and fibers form. In the last stage, the maturation stage, the biomechanical strength of the tendon gradually recovers [4,5,6].

For acute and chronic tendon injuries, different treatment methods are usually adopted in clinical practice (Table 1). However, as there are few blood vessels in the tendon and the metabolism is low, traditional treatment methods can not completely restore the structure and function of the tendon, and may even cause complications such as tendon adhesion and osteolysis [7,8,9]. The rotator cuff is a typical tendon structure composed of tendons from the supraspinatus, infraspinatus, teres minor, and subscapularis muscle, which is prone to injury. It is known that rotator cuff tear (RCT) and anterior cruciate ligament (ACL) injury are the two most common diseases that require tendon graft, which may cause severe pain, dysfunction and even disability [10,11]. After the reconstruction of the rotator cuff and ACL, the rates of re-tear and failure are still high [11,12]. The most important reason may be that the tendon–bone healing process is not ideal [1,13].

The tendon–bone interface, where the tendon or ligament insert into bone, is also called “enthesis” [20]. It includes both soft and hard tissues, which are divided into four gradual and continuous histological zones. The first zone is the tendon, consisting of well-aligned type I collagen fibers, type II collagen, elastin, and small leucine-rich proteoglycan; the second zone is unmineralized fibrocartilage, predominantly consisting of type II and III collagen; the third zone, mineralized fibrocartilage, contains type II and III collagen, as well as aggrecan and type X collagen; the last zone is bone [21,22,23]. Compared with other tissues, there are very limited blood vessels at the tendon–bone interface. As a result, oxygen, growth factors, and nutrients delivered here are insufficient, which may exacerbate the inflammatory reaction and affect cell proliferation and tissue reconstruction [24,25]. It was reported that the key biological process of tendon–bone healing after ACL reconstruction may last for more than 1 year [26]. Therefore, there is no denying that tendon–bone healing is one of the most significant but challenging processes in tendon healing.

In recent years, the application of biomaterials in medicine has become increasingly widespread and it is important to select appropriate biomaterials for the treatment of specific diseases [27,28,29]. In order to promote tendon and tendon–bone healing, many biological methods based on platelet plasma [30], growth and differentiation factors [31], cell therapy [32] and tissue engineering [33] have been used (Table 2). Mesenchymal stem cells (MSCs) can be easily isolated from tissues and have the potential to differentiate into multiple cells. Multiple studies have demonstrated their effectiveness in tendon and tendon–bone healing [32,34,35]. However, stem cell therapy may pose clinical risks such as immune rejection, thrombosis and some ethical problems [36,37,38]. It was also reported that bone marrow mesenchymal stem cells cultured in vitro are prone to malignant transformation spontaneously, which is a serious threat to the safety of cell-based therapies and regenerative medicine [38]. Therefore, cell-free therapies based on stem cell secretions have become the focus of attention. Exosomes, a kind of extracellular vesicle with a diameter of 30~150 nm secreted by cells, which contain a variety of proteins, lipids and nucleic acids such as microRNA (miRNA), mRNA and DNA, play an important role in cell-to-cell communication [39,40,41]. Exosomes have been reported to play an effective therapeutic role in a variety of cancers [42,43,44], neurological diseases [45] and acute organ injuries [46]. In recent years, exosomes have also been proven to promote tendon and tendon–bone healing [8,9,47,48,49,50,51,52,53,54,55,56,57,58,59,60,61,62,63,64,65,66,67,68,69,70,71,72,73,74,75,76,77,78,79,80,81,82,83,84,85,86,87,88,89,90].

The purpose of this systematic review is to summarize and scrutinize existing preclinical studies to illustrate the following aspects: 1. The efficacy of exosomes in tendon healing and tendon–bone healing, and how to improve the therapeutic effect of exosomes. 2. Separation and administration methods of exosomes. 3. The mechanisms of exosomes promoting tendon healing and tendon–bone healing. 4. Inconsistency in existing studies and possible explanations. This is the most up-to-date synthesis of evidence on this topic, and we first systematically reviewed the signaling pathways and gene expression changes of exosomes in promoting tendon healing and tendon–bone healing to elucidate their mechanisms. For different animal models, treatment parameters and some key therapeutic outcomes, we present and discuss the results of tendon healing studies and tendon–bone healing studies separately.

## 2. Materials and Methods

### 2.1. Methods Literature Retrieval

This study was conducted according to the Preferred Reporting Items for Systematic Reviews and Meta-Analyses (PRISMA) statement protocol published in 2020 [97] (the checklist is shown in Appendix A). A flowchart of the literature screening is provided in Figure 1. On 21 January 2023, a systematic and comprehensive search for this review was performed. We used the following key words: “exosome*”, “extracellular vesicle*”, “exosomal”, “Cell-Derived Microparticle*”, “secretome*”, “tendon*”, “ligament*”, “rotator cuff”, “Achilles”, “enthesis”, “footprint”, “tendinous*” “tendinopathy”, “Tendinopathies”, “Tendinosis” and “Tendinitis”. The electronic databases searched included Medline (through PubMed), Web of Science, Embase, Scopus, Cochrane Library and Ovid. The specific search strategy is shown in Appendix A. The methods of analysis, inclusion and exclusion criteria are shown in Appendix A. A “snowball” search was also carried out to identify additional eligible studies by searching the reference lists of articles eligible for a full-text review.

All studies found by our search were imported into Endnote 20, which helped remove the duplicates. The eligibility assessment was conducted independently by MRZ and JZW. Two reviewers completed the title, abstract and conclusion of each article, and excluded the articles whose study types were Patent, Conference Abstract, Letter, Case report, Editorial, or Review and articles focusing on periodontal ligament or dental-related diseases. In case of disagreement, a third reviewer (ZXS) was contacted to discuss whether to include the article. Next, full-text screening of the rest of the articles was conducted independently by the two reviewers based on the inclusion and exclusion criteria shown in Appendix A. Additionally, a third reviewer (ZXS) was contacted to solve any disagreement.

It is significant to point out that current methods for exosome purification are immature and imperfect. Sometimes, other types of extracellular vesicles are present in purified exosomes, resulting in a diameter distribution that is not completely contained within the range of 30~150 nm [47,48,51,61,62,71,74,78,98,99,100]. So, for articles that focus on the effect of extracellular vesicles promoting tendon or tendon–bone healing, we examined the distribution of extracellular vesicle diameters. Articles in which the distribution of diameters was significantly outside the distribution of exosomes (30~150 nm) were excluded [98,99,100]. In this review, “EV”, “exosome” and “extracellular vesicle (EV)” all indicate “exosome”.

### 2.2. Data Extraction

Data extraction was individually conducted by two reviewers. In case of disagreement, the reviewers discussed until an agreement was reached. Apart from the descriptives of individual included studies, other data were collected, such as: 1. Characteristics and isolation of exosomes, which include the origin, source, isolation methods, storage condition, verification methods, size distribution and biomarkers of exosomes. 2. Information of in vivo experimental animal models, which include how the model was established, gender, euthanasia time and number of animals used in each group; the concentration, volume, and delivery route and frequency of the reagent. 3. In vivo therapeutic outcomes, including morphological outcomes, histology outcomes, biomechanical outcomes, and biochemical outcomes. 4. In vitro experimental outcomes, which include cell survival, proliferation, differentiation and migration.

### 2.3. Assessment of Quality of Studies

Assessment of the quality of studies was also conducted by the two reviewers. The Systematic Review Centre for Laboratory Animal Experimentation (SYRCLE) risk of bias assessment tool was used to evaluate the selection bias, performance bias, detection bias, attribution bias, reporting bias and other biases of the studies [101].

### 2.4. Data Analysis

The results of the studies were largely evaluated qualitatively as there were insufficient quantitative data in the studies.

## 3. Results

### 3.1. Search Results

Details about the selection process are shown in a PRISMA flow chart in Figure 1. After searching the six databases, a total of 1794 studies were found. Forty-five articles were included after screening and all of them were case–control studies [8,9,47,48,49,50,51,52,53,54,55,56,57,58,59,60,61,62,63,64,65,66,67,68,69,70,71,72,73,74,75,76,77,78,79,80,81,82,83,84,86,87,88,89,90]. Another article was included by searching the references of the included articles and review articles related to the topic [85]. Finally, 46 studies were included for synthesis.

### 3.2. Assessment of Quality of Studies

The detailed assessment of the quality of each study is demonstrated in Appendix A. All studies had a low risk of baseline characteristics under selection bias, random outcome assessment under detection bias, attrition bias, reporting bias, and other biases [8,9,47,48,49,50,51,52,53,54,55,56,57,58,59,60,61,62,63,64,65,66,67,68,69,70,71,72,73,74,75,76,77,78,79,80,81,82,83,84,85,86,87,88,89,90]. All studies reported baseline characteristics of the animals used which included the species, age and gender of the animals, and the outcomes were randomly assessed [8,9,47,48,49,50,51,52,53,54,55,56,57,58,59,60,61,62,63,64,65,66,67,68,69,70,71,72,73,74,75,76,77,78,79,80,81,82,83,84,85,86,87,88,89,90]. Therefore, all studies were assigned low risk for baseline characteristics and random outcome assessment [8,9,47,48,49,50,51,52,53,54,55,56,57,58,59,60,61,62,63,64,65,66,67,68,69,70,71,72,73,74,75,76,77,78,79,80,81,82,83,84,85,86,87,88,89,90]. All studies reported average results of all animal experiments; therefore, they were assigned low risk for random outcome assessment [8,9,47,48,49,50,51,52,53,54,55,56,57,58,59,60,61,62,63,64,65,66,67,68,69,70,71,72,73,74,75,76,77,78,79,80,81,82,83,84,85,86,87,88,89,90]. No studies reported the death of animals before the end of experiments, which ensured the low risk of attrition bias. No studies reported relevant information on sequence generation and allocation concealment. So, for these two categories, all studies were assigned unclear risks. Ten studies reported that animals were housed in the same and specific living environment, so they had a low risk of random housing [8,9,47,48,50,58,62,66,80,89]. However, one study pointed out that animals were not assigned randomly, and the order of surgical treatments was not randomized either, which would be more economical [59]. So, this study was assigned a high risk of performance bias which included random housing and blinding [59]. Other studies did not report the use of blinding methods during the process of animal housing; therefore, they were assigned unclear risks for blinding under performance bias [8,9,47,48,49,50,51,52,53,54,55,56,57,58,60,61,62,63,64,65,66,67,68,69,70,71,72,73,74,75,76,77,78,79,80,81,82,83,84,85,86,87,88,89,90]. Sixteen studies reported blinding of the assessors when assessing for the outcomes and were therefore classified as low risk for blinding under detection bias [8,47,48,56,57,60,62,69,70,72,73,74,79,81,82,90]. On the whole, the risk of biases in the included studies was low [8,9,47,48,49,50,51,52,53,54,55,56,57,58,59,60,61,62,63,64,65,66,67,68,69,70,71,72,73,74,75,76,77,78,79,80,81,82,83,84,85,86,87,88,89,90].

### 3.3. Source of Exosomes

The detailed results of sources of exosomes were shown in Appendix A. Human MSC exosomes [8,9,47,48,51,55,58,60,63,64,65,69,70,74,76,79,83,90] and rat MSC exosomes [49,53,62,66,67,71,72,73,75,77,78,81,82,85,88,89] were the two most common sources, and about one-third of studies used them, respectively. Mouse MSC exosomes [50,54,56,68,80,84] and rabbit MSC exosomes [57,61,86] were also used in a few studies. Three studies used purified exosome products [52,59,87]. Sources of human MSC exosomes included adipose MSCs [60,65,74,76,79,90], umbilical cord MSCs [9,55,63,69,70], induced pluripotent MSCs [47,51,58], bone marrow MSCs [48,64,83] and tendon stem/progenitor cell (TSPCs) [8]. Sources of rat MSC exosomes included bone marrow MSCs [62,71,73,82,85,88,89], adipose MSCs [72,75,77,78], tendon stem cells (TSCs) [49,53,66,67] and infrapatellar fat pad MSCs [81]. Sources of mouse MSC exosomes included bone marrow MSCs [80,84], bone-marrow-derived macrophages (BMDMs) [56,68], fibroadipogenic progenitor (FAPs) [54], and adipose MSCs [50]. Sources of rabbit MSC exosomes included bone marrow MSCs [57,86] and adipose MSCs [61].

In most studies, the origin and source of exosomes across experimental groups were generally the same. However, in one study, Hayashi et al. compared the therapeutic effect of EVs derived from bone marrow MSCs at passage 5 and passage 12. The results showed that mice treated with P_5_ EVs demonstrated a better therapeutic effect. There were no significant adhesions between the tendon and surrounding tissue, the histological score was better, and the newly formed collagen fibers and tendon were similar to normal tendon tissue. However, different from P_12_ EVs, the particle size diameter range of P_5_ EVs was contained within the range of exosome (30~150 nm) [48].

### 3.4. Isolation and Characterization of Exosomes

The detailed results of the isolation and characterization of exosomes were shown in Appendix A. Ninety-one percent of studies isolated exosomes based on centrifugation [8,9,47,48,49,50,51,53,54,55,56,57,58,60,61,62,63,64,65,66,67,68,69,70,71,72,73,74,75,76,77,78,79,80,81,82,83,84,85,88,89,90], among which thirty-eight studies performed differential centrifugation followed by ultracentrifugation [8,9,47,48,49,50,51,53,54,55,57,58,60,62,63,64,65,66,67,69,71,72,73,74,75,76,77,78,79,80,81,82,83,84,85,88,89,90]. Three studies used purified exosome products. [52,59,87] One study used a total exosome isolation reagent to obtain exosomes [86]. As for characterization, nanoparticle tracking analysis, transmission electron microscope and Western blot were the three most common methods to visualize exosomes and analyze their size distribution. Other methods such as nano-flow analysis (NFA), dynamic Light Scattering (DLS), flow cytometry (FCM), atomic force microscopy (AFM) and colloidal nano plasmonic assay (CONAN) were also used by a few studies [8,47,62,64,70,78,86]. The size distribution of exosomes mainly ranged from approximately 30 to 150 nm. However, as we mentioned before, as current methods for exosomes purification are immature and imperfect, the exosomes isolated were sometimes not completely pure [9,47,48,51,53,61,62,66,69,71,73,74,78,83,85,86,87]. Interestingly, two studies compared the therapeutic effect of exosomes with extracellular vesicles with larger diameters and they reported different results [58,72]. Ye et al. reported that the therapeutic effects of exosomes and EVs were similar while Xu et al. pointed out that exosomes demonstrated better therapeutic effects [58,72]. CD9, CD63, CD80, TSG101 and HSP70 were the most common positive makers of exosomes. Some studies also reported the absence of negative makers of exosomes including GM130 and Calnexin [47,51,58,63,65,69,72,74,83,85,86,89,90]. Fifteen studies reported that the exosomes were stored at −80 °C before being used [9,47,55,58,61,62,70,72,73,74,83,84,85,87,88], while one study stored exosomes at 4 °C [54].

### 3.5. Animal Models

The detailed results of animal models were shown in Table 3 and Table 4. Among the thirty-five studies on tendon healing [8,9,47,48,49,50,51,52,53,54,55,56,57,58,59,60,61,62,63,64,65,66,67,68,69,70,71,72,73,74,75,76,77,78,79], five animal species were used to construct animal models, which included rats, mice, rabbits, dogs and sheep. Most studies chose male animals. Rats were the most common animal used for animal models, and were used in twenty-two studies [8,9,47,49,51,53,55,58,62,63,65,66,67,69,71,72,73,74,75,76,77,78], and Sprague Dawley rats were used in twenty studies, which were the most frequent [8,9,47,49,51,55,58,63,65,66,67,69,71,72,73,74,75,76,77,78]. One-fifth of the studies used mice [48,50,54,56,64,68,79], and New Zealand rabbits [57,59,60,61], dogs [52], sheep [70] were also used by a few studies. Methods for constructing tendon injury models can be divided into three main categories: performing surgeries; injection of solution; training animals with a treadmill. Eighty percent of studies performed surgeries on animals [8,9,48,49,50,52,54,55,56,57,59,60,61,62,63,64,65,67,68,69,70,71,73,74,75,76,77,78]. As the Achilles tendon and rotator cuff were the most common sites of tendon injury, they were damaged for animal models in twelve [9,48,49,50,55,59,61,62,63,64,69,74] and seven [54,57,60,65,70,75,76] studies, respectively. The central part of the patellar tendon [8,67,71,73,77], the flexor digitorum longus [56,68], and the tendon of the interphalangeal joint [52] were resected or removed to establish animal models in a few studies. Six studies injected solutions to create models [47,51,53,58,66,72]. Three injected a carrageenan solution [47,51,58], and three injected a type I collagenase solution [53,66,72]. One study used a treadmill to train the mice, which is a novel method and may simulate the natural degeneration process of the tendon [79].

Across the 11 studies on tendon–bone healing [80,81,82,83,84,85,86,87,88,89,90], Sprague Dawley rats were used to construct animal models in nearly half of the studies [81,82,83,87,88]. Three studies used C57BL/6J mice [80,84,85], two used rabbits [86,90] and one used Wistar rats [89]. Similar to tendon healing studies, most animals were male. However, the tendon site and specific method of constructing animal models of tendon–bone healing studies are significantly different from those of tendon healing studies. The majority of studies resected the anterior cruciate ligament and created bone tunnels [81,82,83,89] or constructed tendon–bone injury models at the rotator cuff [85,86,87,88,90]. Only two studies damaged the enthesis of the Achilles tendon [80,84].

### 3.6. Group Assignment and Treatment Parameters of Animal Experiments

The detailed results of treatment parameters are shown in Appendix A. Different concentrations of exosomes were used in different studies. Among all 46 studies [8,9,47,48,49,50,51,52,53,54,55,56,57,58,59,60,61,62,63,64,65,66,67,68,69,70,71,72,73,74,75,76,77,78,79,90], only 1 compared the effect of the concentration of exosomes on therapeutic efficacy [62]. Gissi et al. compared between 2.8 $\times 10^{12}$ particles/mL and 8.4 $\times 10^{12}$ particles/mL of exosomes. The studies reported that exosomes of high concentration (8.4 $\times 10^{12}$ particles/mL) significantly promoted the expression of type I collagen fibers and inhibited the expression of type III collagen fibers, with higher histological scores and more obvious promotion of tendon healing [62]. However, even for this study, it only set up two different concentrations for comparison. There were no studies that set up multiple different concentration gradients to explore the most appropriate concentration for in vivo administration [8,9,47,48,49,50,51,52,53,54,55,56,57,58,59,60,61,62,63,64,65,66,67,68,69,70,71,72,73,74,75,76,77,78,79,90].

Direct injection and implantation with biomaterials are the two most common ways to deliver exosomes. Thirty-five studies used different injection methods, which included local injection, subcutaneous injection and intravenous injection [9,47,48,49,51,53,54,56,58,60,61,62,63,64,65,66,67,69,71,72,73,75,76,78,79,80,81,82,83,84,85,86,88,89,90]. Different from studies on tendon healing, whose injection site was the tendon, the injection sites of the studies on tendon–bone healing included a bone tunnel, joint cavity or the tendon–bone interface [9,47,48,49,51,53,54,56,58,60,61,62,63,64,65,66,67,69,71,72,73,75,76,78,79,80,81,82,83,84,85,86,88,89,90]. Ten studies used biomaterials which included collagen sheet, hydrogel, fibrin glue, fiber patch, type I collagen scaffold, collagen sponge and fibrin sealant to carry exosomes [8,50,52,55,57,59,70,74,77,87]. Although these delivery methods based on biomaterials were proven to be therapeutically effective and control groups in which animals were only treated with the biomaterials were used, no study used an additional group of animals only treated with exosomes to explore whether the biomaterials promote or inhibit the therapeutic efficacy of exosomes [8,50,52,55,57,59,70,74,77,87].

As for frequency of administration, only six studies conducted administration multiple times, and the frequencies were weekly, at day 1 and day 7, once every three days, twice a week, weekly, and day 0, day 3 and day 7, respectively [47,48,56,66,81,89]. Different from other studies, administration was initiated several weeks after model establishment in six studies to establish chronic injury animal models [57,75,76,79,88,90]. The optimal timing and frequency of administration in vivo remain to be investigated, as no studies compared the effects of different dosing times and dosing frequencies on the test results.

### 3.7. Methods of Modification to Improve the Biological Function of Exosomes

To enhance the biological function of exosomes, many strategies were used to precondition the MSCs or exosomes. Ten studies preconditioned MSCs before the isolation of exosomes [50,54,63,68,71,74,80,82,83,85]: Shen et al. preconditioned adipose MSCs with interferon-γ (IFN-γ) primers, Davies et al. compared the therapeutic effect of exosomes from FB (FAPs that have assumed a beige adipose tissue differentiation state) with exosomes from NFB (FAPs that did not assumed a beige adipose tissue differentiation state), Li et al. preconditioned human umbilical cord MSCs with hydroxycamptothecin, Yu et al. down-regulated circRNA-Ep400 of macrophages before the isolation of exosomes, Li et al. preconditioned bone marrow MSCs with Eugenol, Xu et al. preconditioned adipose MSCs with bioactive glasses, Wang et al. used bone marrow MSCs overexpressing Scleraxis (Scx) to isolate exosomes, Zhang et al. simulated the MSCs under hypoxia circumstance, Wu et al. used magnetically actuated MSCs, and Wu et al. preconditioned bone marrow MSCs with Low-Intensity Pulsed Ultrasound Stimulation (LIPUS) [50,54,63,68,71,74,80,82,83,85]. Three studies preconditioned exosomes before in vivo administration [53,86,89]: Liu et al. modified exosomes by a nitric oxide nanomotor, Han et al. treated exosomes with BMP-2-loaded microcapsules, and Li et al. overexpressed miR-23a-3p in bone marrow MSC exosomes [53,86,89]. All the studies above reported that pre-treated exosomes or exosomes secreted by pre-treated MSCs demonstrated better therapeutic effects, which meant the modifications were effective [50,53,54,63,68,71,74,80,82,83,85,86,89].

### 3.8. Histological Outcomes

The detailed results of key therapeutic outcomes were shown in Table 5 and Table 6 on tendon healing, the therapeutic effects of exosomes on different animal models were similar, while there were still some differences [8,9,47,48,49,50,51,52,53,54,55,56,57,58,59,60,61,62,63,64,65,66,67,68,69,70,71,72,73,74,75,76,77,78,79]. In Achilles tendon injury models, animals treated with exosomes demonstrated higher fiber expression, higher histological scores, and collagen type I/III ratios close to that of normal tendons [9,48,49,50,55,59,61,62,63,64,69,74]. In addition, exosomes significantly inhibited the inflammatory reaction, inhibited the adhesion of tendon and surrounding tissues, promoted the maturation of blood vessels, and made the arrangement of fibers and blood vessels more orderly [9,48,49,50,55,59,61,62,63,64,69,74]. In patellar tendon injury models, exosomes increased the number of TSCs, increased the expression and deposition of type I collagen, and the density and arrangement of cells in the injured area were more similar to those of normal tendons [8,67,71,73,77]. The expression of type III collagen was promoted in the early stage while inhibited in the late stage [67]. In rotator cuff injury models, the treatment of exosomes significantly inhibited fat infiltration, promoted the formation of fibrocartilage, and made collagen fibers arrange more orderly [54,57,60,65,70,75,76]. The expression of type I collagen was promoted and the expression of type III collagen was suppressed [60]. In tendon injury models induced by carrageenan or collagenase, exosomes reduced the infiltration of inflammatory cells and inflammatory reaction, promoted the expression of type I collagen and the formation of the extracellular matrix (ECM), and inhibited the expression of type III collagen [47,51,53,58,66,72]. The remaining five studies reported that exosomes promoted the expression of type I collagen, inhibited the expression of type III collagen, promoted the proliferation and migration of fibroblasts and tenocytes, and induced peritendon fibrosis [52,56,68,78,79].

Three kinds of animal models—ACL injury, rotator cuff injury and Achilles tendon injury—were established in eleven tendon–bone healing studies [80,81,82,83,84,85,86,87,88,89,90]. In the ACL injury models, animals treated with exosomes demonstrated higher histological scores, higher expression of fibers and cartilage, lighter inflammatory reaction and better angiogenesis [81,82,83,89]. In the rotator cuff injury models, exosomes significantly inhibited fat infiltration, promoted the expression of fibrocartilage and type I collagen, and promoted angiogenesis, thus promoting the formation of ECM and accelerating the process of tendon–bone healing [85,86,87,88,90]. In contrast to the control group, which showed severe fatty infiltration and collagen disorder, the treatment of exosomes was effective [85,86,87,88,90]. In the Achilles tendon injury models, exosomes inhibited osteoclastogenesis and prevented osteolysis [80,84]. As the number of chondrocytes increased, collagen became arranged in order, and the transitional structure of the tendon–bone interface gradually formed [80,84].

On the whole, exosomes promoted the expression of type I collagen and fibrocartilage and optimized the arrangement of fibers and blood vessels, thus promoting the formation of ECM [8,9,47,48,49,50,51,52,53,54,55,56,57,58,59,60,61,62,63,64,65,66,67,68,69,70,71,72,73,74,75,76,77,78,79,80,81,82,83,84,85,86,87,88,89,90]. At the same time, exosomes inhibited the expression of type III collagen and osteolysis and inhibited inflammatory reaction and fat infiltration, thus promoting tendon healing and tendon–bone healing [8,9,47,48,49,50,51,52,53,54,55,56,57,58,59,60,61,62,63,64,65,66,67,68,69,70,71,72,73,74,75,76,77,78,79,80,81,82,83,84,85,86,87,88,89,90]. However, the roles of exosomes in inhibiting tendon adhesion and promoting peritendinous fibrosis were controversial in several studies, which are worth discussing [9,56,63,68].

### 3.9. Biomechanical Outcomes

Across the 11 studies on tendon–bone healing [80,81,82,83,84,85,86,87,88,89,90], biomechanical outcomes were reported in 10 studies, which all showed positive effects on biomechanical properties [80,81,82,83,84,85,87,88,89,90]. The biomechanical test indexes included stiffness, ultimate failure load, Young’s modulus and stress [80,81,82,83,84,85,86,87,88,89,90].

Similar to studies on tendon–bone healing, sixty-nine percent of studies on tendon healing reported biomechanical outcomes and the indexes also included stiffness, ultimate failure load, Young’s modulus and stress [8,9,47,51,52,55,56,57,59,60,61,63,64,65,66,67,68,72,73,74,75,76,77,79]. Interestingly, two studies used SWB (static weight-bearing) and PWT (paw-withdrawal threshold) to measure the extent of pain, and both of them reported that the SWB and PWT of groups treated with exosomes were obviously improved, which indicated that the pain induced by tendinopathy was significantly relived [47,51]. However, there were four studies that did not report obvious positive effects [9,56,59,63]. Yao et al. and Li et al. both studied the effect of exosomes on tendon adhesion, both of which reached the conclusion that there was not an obvious difference in biomechanical properties between groups [9,63]. Additionally, Cui et al. reported that the biomechanical properties of the group treated with exosomes and the control group showed few differences [56]. Wellings et al. reported that the promoting effect was not obvious [59]. Chamberlain et al. reported that even though exosomes promoted the biomechanical properties of the tendon, the effect of exosome-educated macrophages was better [64]. From the results reported in 34 articles [8,9,47,51,52,55,56,57,59,60,61,63,64,65,66,67,68,72,73,74,75,76,77,79,80,81,82,83,84,85,87,88,89,90], the role of exosomes in enhancing the biomechanical properties of tendons or tendon–bone interfaces is indisputable, but there are still some issues worth discussing.

### 3.10. Macroscopic Appearance and Morphological Outcomes

Some studies on tendon healing reported the macroscopic appearance of experimental animals, but most of the outcomes were limited to visual observation [9,53,56,60,63,67,70,72,73,74]. On the whole, the injured tendons treated with exosomes demonstrated better healing [9,53,56,60,63,67,70,72,73,74]. To be specific, Liu et al. reported that the inflammation areas were significantly decreased with the treatment of exosomes [53]. Yao et al., Li et al. and Jenner et al. reported that exosomes inhibited the adhesion of tendons and surrounding tissues effectively [9,63]. Wang et al. reported that the group treated with exosomes was similar to the sham group macroscopically, with a significant increase in the thickness of the supraspinatus tendon [60]. Xu et al. reported that the treatment of EV_B_ reduced scarring and deleterious morphological changes of the Achilles tendon [74]. Cui et al. reported that exosomes promote the formation of fibrotic tissue [56]. Xu et al. and Jenner et al. confirmed that exosomes reduced the inflammatory reaction with the help of nuclear magnetic resonance imaging [70,72].

Six out of eleven studies of tendon–bone healing reported the macroscopic appearance and morphological outcomes [81,82,83,86,88,89]. Similarly, the macroscopic appearances of the tendon–bone interface of the groups treated with exosomes were better, as there was more fibrocartilage and collagen fibers were more orderly arranged [81,82,83,86,88,89]. The morphological analyses were mainly conducted by micro-CT [81,82,83,86,89], and two studies used X-ray to analyze in addition [83,89]. According to the outcomes of micro-CT and X-ray, the animals treated with exosomes showed smaller mean bone tunnel areas and higher BV/TV ratios, which provided strong evidence for exosomes promoting tendon–bone healing [81,82,83,86,89]. Additionally, Wu et al. reported that the trabecular thickness, trabecular number, trabecular separation, structure model index, and bone mineral density of groups treated with exosomes were significantly improved [83].

### 3.11. Macrophage Polarization and Regulation of Inflammatory Reaction

Twenty-three out of forty-six studies reported that exosomes significantly inhibited the inflammatory reaction, down-regulated the expression of pro-inflammatory cytokines and M1 macrophage markers which included TNF-α, IL-1β, IL-6, IL-8, CD31, CD86, NGF, CCR7, COX-2 and NOS-2, up-regulated the expression of anti-inflammatory cytokines and M2 macrophage markers which included CD163, CD206, IL-10 and TGF-β, and promoted M2 polarization of macrophages [47,49,50,51,52,53,58,61,64,65,74,75,76,77,78,79,81,84,86,87,88,89,90]. In terms of mechanism, Ye et al. reported that exosomes delivered DUSP2 and DUSP3 to macrophages and inhibited the activation of the P38 MAPK signaling pathway, thus promoting M2 polarization of macrophages [58].

### 3.12. MicroRNAs and Signaling Pathways

Multiple microRNAs and signaling pathways were reported to play important roles in tendon and tendon–bone healing. Across the 35 studies on tendon healing [8,9,47,48,49,50,51,52,53,54,55,56,57,58,59,60,61,62,63,64,65,66,67,68,69,70,71,72,73,74,75,76,77,78,79], fourteen studies reported nine microRNAs and nine signaling pathways which were suggested to be important intermediate processes in the mechanisms of exosomes promoting tendon healing [8,9,49,51,55,56,58,63,67,68,69,74,77,78]. For instance, Zhang et al. reported that the activation of PI3K/AKT and MAPK/ERK1/2 signaling molecules might play an important role in tendon healing [49]. Tao et al. reported that with the treatment of exosomes, H19 regulated YAP phosphorylation and translocation through H19-pp1-YAP interaction, thus promoting tendon-related gene expression [8]. Gao et al. reported that exosomes inhibited the activation of mast cells via the HIF-1 signaling pathway and reduced pain induced by tendinopathy [51]. Yao et al. reported that HUMSC exosomes might manipulate p65 activity by delivering low-abundance miR-21a-3p, ultimately inhibiting tendon adhesion [9]. Cui et al. reported that miR-21-5p from exosomes acted on Smad7 to promote the proliferation and migration of tendon cells and fibroblasts [56]. Yao et al. reported that exosomes regulated the PTEN/mTOR/TGF-β1 pathway via miR-29a-3p, up-regulated the expression of tendon-related genes, promoted TSCs differentiation into the tendon, and thus promoted tendon healing [55]. Ye et al. reported that exosomes delivered DUSP2 and DUSP3 to macrophages and inhibited the activation of the P38 MAPK signaling pathway, thus promoting M2 polarization of macrophages [58]. Li et al. reported that HCPT-EVs up-regulated the expression of GRP78, CHOP and BAX, and down-regulated the expression of BCL-2, which might activate the ERS pathway to inhibit adhesion [63]. Song et al. reported that exosomes regulated cell proliferation, migration and tendon healing via mi-144-3 [67]. Yu et al. reported that exosomes containing circ-RNA-Ep400 secreted by M2 macrophages promoted peritendinous fibrosis via the miR-15b-5p/FGF-1/7/9 pathway [68]. Han et al. reported that HUMSC exosomes promoted tendon healing by down-regulating ARHGAP5 expression and activating RhoA via mir-27b-3p [69]. Xu et al. reported that HSA-miR-125a-5p, HSA-miR-199b-3p, and miR-92b-5p were highly expressed in EV_B_, and the first two regulated macrophage polarization and promoted angiogenesis [74]. Liu et al. reported that exosomes promoted TSC proliferation and migration by activating Smad2/3 and Smad1/5/9 signaling pathways [77]. Zhao et al. reported that exosomes inhibited IGFBP3 expression via miR-19a, promoted tendon cell proliferation and reduced apoptosis rate [78].

Across the eleven studies on tendon–bone healing [80,81,82,83,84,85,86,87,88,89,90], five studies reported the roles of micro RNAs and signaling pathways [80,83,85,88,89]. Huang et al. reported that rat bone marrow MSC exosomes activated the Hippo signaling pathway through VEGF, which promoted the proliferation and migration of HUVECs (human umbilical vein endothelial cells) and thus promoted tendon–bone healing [88]. Wang et al. reported that exosomes secreted by Scx-overexpressing bone marrow MSCs targeted OCSTAMP and CXCL12 via miR-6924-5p, which inhibited osteolysis and thus promoted tendon–bone healing [80]. Li et al. reported that bone-marrow MSCs exosomes promoted M2 polarization of macrophages via miR-23a-3p [89]. Wu et al. reported that IONP-exosomes (exosomes derived from magnetically actuated bone morrow mesenchymal stem cells) promoted tendon–bone healing by down-regulating SMAD7 via miR-21-5p, promoting NIH3T3 fibroblast proliferation and migration, and up-regulating the expression of COL I, COL III, and α-SMA [83]. Wu et al. reported that exosomes from bone marrow MSCs preconditioned by LIPUS promoted tendon–bone healing by up-regulating the expression of chondrogenic genes and down-regulating the expression of adipogenic genes via miR-140 [85].

In summary, exosomes promoted the expression of angiogenesis, cell proliferation, migration and differentiation, macrophage M2 polarization, chondrogenesis and collagen production via multiple microRNAs and signaling pathways, effectively promoting tendon healing and tendon–bone healing. As we know, exosomes contain a variety of proteins, RNAs, and growth factors [39,40,41]. So, these reported results are not surprising.

### 3.13. Changes in Gene Expression

Twenty-five studies reported that the treatment of exosomes altered the expression of multiple genes [9,49,50,52,53,55,56,61,63,67,68,71,73,74,75,76,77,78,79,80,83,84,85,86,87]. Expressions of several types of genes were up-regulated, including the type I collagen gene; the inhibitor of metalloproteinase gene, TIMP-1; chondrogenic genes, COL II and SOX-9; tenogenesis genes, TNMD, TNC, Scx, DCN and MKX; ECM genes, BGN and ACAN; fibroblast growth factor genes, FGF-1, FGF-7 and FGF-9; the cell proliferation gene, PCNA; osteogenic genes, RUNX2 and OCN; and anti-inflammatory genes, IL-10, IGF-1, IGF-2 and TGF–β [9,49,50,52,53,55,56,61,63,67,68,71,73,74,75,76,77,78,79,80,83,84,85,86,87]. Additionally, expressions of some genes were down-regulated, which included the type III collagen gene; metalloproteinase gene, MMP; proinflammatory genes, IL-6, TNF-α, IL-1α and IL-1β; proapoptotic genes, Caspase-3 and IGFBP3; osteoclastogenic genes, ACP5, CALCR, NFATc1 and ITGB3; and adipogenic genes, Adipo, Retn and Pparg [9,49,50,52,53,55,56,61,63,67,68,71,73,74,75,76,77,78,79,80,83,84,85,86,87].

Interestingly, four studies reported that the expression of α-SMA was promoted [56,68,83], while three other studies reported the opposite conclusion [9,49,63]. This paradox may be related to proper fibrosis and excessive scarring, which will be discussed in the “Discussion” section.

### 3.14. In Vitro Experiment Outcomes (Cell Proliferation, Migration and Differentiation)

Some studies also provided results of in vitro cell experiments to further verify the conclusion of in vivo experiments [8,9,47,48,49,53,54,55,56,57,61,62,63,67,68,69,71,73,74,75,76,77,78,82,83,87,88]. Twenty-five studies reported that exosomes had a positive effect on the proliferation or/and migration or/and differentiation of a variety of cells which included tenocytes, TPSCs, TSCs, fibroblast, endothelial cells, HUVECs, bone marrow MSCs and osteoblasts [8,47,48,49,53,54,55,56,57,61,62,67,68,69,71,73,74,75,76,77,78,82,83,87,88]. Among these, five studies found that this positive effect was in a dose-dependent manner [49,51,61,69,76]. Ren et al. reported that purified exosome products significantly promoted the proliferation and migration of tenocytes and osteoblasts and accelerated the fusion of the two cells [87]. This was the only study that focused on the effect of exosomes on osteoblasts, which provided strong evidence that exosomes promoted tendon–bone healing [87]. Strangely, two studies about exosomes inhibiting tendon adhesion reported that exosomes inhibited the proliferation of fibroblasts, which deserves to be discussed [9,63]. On the whole, exosomes significantly promoted cell proliferation, migration and differentiation, and thus promoted tendon and tendon–bone healing.

## 4. Discussion

From the above results, it can be seen that exosomes can effectively promote tendon healing and tendon–bone healing, and their therapeutic efficacy is mainly manifested in the following ways: 1. Promoting cell proliferation and differentiation into tendon cells and chondrocytes. 2. Alleviating inflammatory reactions and providing a good microenvironment for tissue repair. 3. Promoting the expression of type I collagen and inhibiting the expression of type III collagen, thereby promoting tissue repair and reducing scar formation. 4. Improving the biomechanical properties of tendons and increasing the strength of the tendon–bone interface. In the following sections, we will mainly discuss: 1. The methods of improving the therapeutic efficacy of exosomes. 2. Separation and administration methods of exosomes. 3. Mechanisms of exosomes promoting tendon and bone healing. 4. Inconsistencies in existing studies and possible explanations.

### 4.1. The Most Suitable Source of Exosomes for Tendon and Tendon–Bone Healing

Across the forty-six studies, the exosomes used were from four species—humans, rats, mice and rabbits—and three studies used purified exosome products [8,9,47,48,49,50,51,52,53,54,55,56,57,58,59,60,61,62,63,64,65,66,67,68,69,70,71,72,73,74,75,76,77,78,79,80,81,82,83,84,85,86,87,88,89,90]. As for the types of cells (most of which are MSCs), nine types were used, which included ASCs, IP-MSCs, BM-MSCs, TSPCs, HU-MSCs, TSCs, IPFP-MSCs, FAPs, and BMDMs. Although exosomes from different sources were proven effective, it is still important to discuss exosomes from which species and which kinds of cells are the most suitable. As obtaining MSCs from human tissue is convenient and rats are the most common experimental animals, it is noticeable that humans and rats are the two most common species, and BM-MSCs and ASCs are the two most common types of MSCs. Different kinds of MSCs have their own characteristics. For example, ASCs are abundant and easy to obtain [102]; TSCs have greater tenogenesis and proliferation capacity [103]; HU-MSCs cost less and take less time to obtain [104]. Martinez-Lorenzo et al. reported that MSCs from rabbits and sheep tissue demonstrated more capacity to differentiate into cartilage than human MSCs [105]. As the formation of fibrocartilage is essential in the process of tendon–bone healing, exosomes from rabbits or sheep MSCs may be better. However, no studies compare exosomes from different species and MSCs, and which kinds of MSCs are the most suitable for extraction of exosomes to treat injured tendons remains to be studied. Hayashi et al. reported that the therapeutic effects of EVs derived from bone marrow MSCs at passage 5 were better than those at passage 12 [48], which poses a new research direction: the appropriate passage number of MSCs used to obtain exosomes.

### 4.2. Modification of Exosomes or MSCs

Multiple studies proved that the efficacy of exosomes was affected by the culture conditions of MSCs and the modification of exosomes. Preconditioning MSCs with small molecules such as IFN-γ primers, hydroxycamptothecin and eugenol was proven to enhance the effect of exosomes [50,63,71]. Special culture conditions such as hypoxia circumstance, magnetic actuation and LIPUS made a positive influence on the potency of exosomes [82,83,85]. Modification of exosomes included preconditioning with small molecules, overexpressing certain factors or microRNAs and loading certain proteins, such as NO, BMP-2 and miR-23a-3p and circRNA-Ep400, which were also effective [53,68,86,89].

We speculate that preconditioning and special culture conditions promote the effect of exosomes mainly in the following two ways: 1. Promoting MSC differentiation in different directions and the expression of genes for tendon healing. 2. Changing the content of non-coding RNA and some functional proteins in exosomes, which, to a large extent, mediate the biological function of exosomes.

There is no denying that some special culture conditions such as hypoxia and low pH improve the production or potency of exosomes [106,107]. However, these extreme-culture conditions are not suitable for the mass production of exosomes, so new culture conditions remain to be explored. MicroRNA, small molecules and certain factors provide future research directions for studying the specific mechanism of exosomes promoting tendon and tendon–bone healing, which are worthy of attention.

### 4.3. Exosomes Isolation and Administration

In the included studies, the most common methods to isolate exosomes were differential centrifugation and ultracentrifugation. Ultracentrifugation is currently the gold standard for the isolation of exosomes, but it also has certain drawbacks: the time of the whole process is long and the purity of exosomes is low [108]. In addition to ultracentrifugation and differential centrifugation, which is also called gradient centrifugation, other methods for isolation include co-precipitation, size-exclusion chromatography, immunoaffinity capture and field flow fractionation [108,109,110,111]. Aside from NFA, DLS, NTA, TEM and AFM, which were reported in the included studies, methods to measure the physical characteristics also include scanning electron microscopy, cryo-electron microscopy, tunable resistive pulse sensing and single EV analysis [112,113,114,115].

It is well acknowledged that current methods for exosome purification are immature and imperfect, and the exosomes isolated were sometimes not completely pure. Additionally, there are some differences between the biogenesis of exosomes and other extracellular vesicles. The volume of microcapsules is relatively large, and they are vesicular structures formed directly through cell germination. The volume of exosomes is small, and their formation process is relatively complex. Usually, cytoplasmic membranes sprout inward to form endosomes that encapsulate proteins and nucleic acids. Subsequently, multiple endosomes approach each other to form larger vesicles, known as a multivesicular body. Finally, the multivesicular body fuse with the plasma membrane and the released vesicles are exosomes. [116]. So, for tendon and tendon–bone healing, they may demonstrate different effects. As we mentioned before, Ye et al. reported that the therapeutic effects of exosomes and EVs were similar, while Xu et al. pointed out that exosomes demonstrated better therapeutic effects than ectosomes [58,72]. However, those comparisons are not convincing enough. More research is needed in this area.

In the included studies, injection was the most common method for the administration of exosomes, which included local injection, subcutaneous injection and intravenous injection. Obviously, with the flow of blood, exosomes could not maintain an effective concentration at the injured site for a long time, so it is not the best administration method. Ten studies used biomaterials which included collagen sheet, hydrogel, fibrin glue, fiber patch, type I collagen scaffold, collagen sponge and fibrin sealant to carry exosomes [8,50,52,55,57,59,70,74,77,87]. Compared with injection, administrating exosomes with biomaterials has some advantages. For example, Shi et al. reported that Tisseel plus a purified exosome product could release exosomes stably for over two weeks [52]. Additionally, hydrogel was also reported to prolong the release time of exosomes and thus improve their bioavailability and therapeutic efficacy [73]. In this way, even with a lower exosome concentration than in other studies, the therapeutic effects were still achieved [73]. Furthermore, a fiber-aligned patch could further promote tendon and tendon–bone healing by providing bioactive stimulation and mechanical support [57]. However, these studies only compare the groups treated with exosomes with biomaterials and groups only treated with biomaterials to prove the role of exosomes. An extra kind of group only treated with exosomes is also needed to demonstrate whether the biomaterials promote or inhibit the efficacy of exosomes. In addition, the promoting effects of different biomaterials on the exosomes also need to be compared. Choosing the most suitable biomaterial for the treatment of a tendon injury is extremely important.

The concentration of exosomes and the frequency of administration also varied across different studies. Although Gissi et al. reported that high-concentration exosomes showed better effects than low-concentration exosomes [62], the most appropriate concentration and frequency of administration still remain to be explored. Different concentration gradients of exosomes were set for in vitro experiments in some studies, which were also needed for in vivo experiments. Another important issue is that as exosomes contain a variety of bioactive components [39,40,41], exosomes are fragile in the external environment and have fragile biological activity. However, no studies monitored the biological activity of exosomes after administration, nor did studies explore the pharmacokinetics of exosomes in vivo after administration. In future research, exosomes with different concentration gradients can be applied to cells or animal models, and their content changes should be monitored over a certain period of time. We can further evaluate the efficacy of exosomes through pharmacokinetic results.

### 4.4. Animal Models

Across studies on tendon healing, Achilles tendon injury models were the most common, while ACL injury models and rotator cuff injury models were the most common in tendon–bone healing studies. This is reasonable as ACL and rotator cuff injuries are usually accompanied by the separation of tendon and bone. In six studies, administration was initiated several weeks after model establishment to establish chronic injury animal models [57,75,76,79,88,90]. Other studies simulated acute tendon injury. However, many chronic tendon injuries in daily life are natural degenerative processes, especially for the elderly. Among the studies included, only one study used a treadmill to train animals to simulate this process [79]. Therefore, the role of exosomes in alleviating tendon degeneration needs further research. It is significant to point out that the animals used for in vivo experiments are relatively young. However, the incidence rate of tendinopathy-like rotator cuff tears is higher in the elderly [90]. More importantly, except for one study which chose sheep to establish animal models [70], animal models in other studies were established on small animals. The problem may be that the healing time of tendon injury in rats is relatively small, which could not completely reflect the healing process of tendon injuries in humans [81]. Therefore, the experiment also needs to be verified in larger animals.

Common biomechanical test indexes included stiffness, ultimate failure load, Young’s modulus and stress. Only two studies used SWB and PWT to measure the extent of pain [47,51]. The evaluation of pain could further evaluate the treatment effect, which is worth carrying out. So, in future studies, the degree of pain could be evaluated before killing animals and taking tendons for biomechanical tests.

### 4.5. Mechanism of Exosomes Promoting Tendon Healing

Although no studies reported the exact mechanisms through which exosomes played their therapeutic roles in tendon and tendon–bone healing, we could still obtain some insights from a large number of reported microRNA, signaling pathways and changes of phenotypes in cells and animals.

The mechanism of exosomes promoting tendon healing can be summarized in the following three ways: 1. Inhibiting inflammatory reaction and regulating macrophage polarization. 2. Promoting the migration and proliferation of tenocytes, up-regulating the expression of collagen fibers, regulating the ratio of type I/III collagen and reducing scar formation. 3. Promoting the expression of tenogenesis factors and cytokines, reconstructing the ECM.

Firstly, exosomes could significantly inhibit the inflammatory reaction and promote the M2 polarization of macrophages. It was reported that M1 macrophages could eradicate harmful bacteria and simulate inflammatory reactions [117]. In general, the number of M1 macrophages increases significantly at the early stage of tendon injury [118]. TNF-α and IL-6 are common makers of M1 macrophages, which are closely related to the NF-κB pathway, and have strong pro-inflammatory effects [119,120]. M2 macrophages could secrete IL-10, a kind of anti-inflammatory factor to inhibit inflammatory reactions [121]. In addition, M2 macrophages could secrete TGF-β and VEGF, which promote tissue repair and the formation of ECM [121,122]. Therefore, M1 macrophages are not conducive to tendon healing, while M2 macrophages promote tendon healing. A number of included studies reported that exosomes down-regulated the expression of M1 macrophage makers and pro-inflammatory factors, up-regulated the expression of M2 macrophage makers and anti-inflammatory factors, promoted M2 polarization of macrophages and significantly inhibited the inflammatory reaction [47,49,50,51,52,53,58,61,64,65,74,75,76,77,78,79]. As for how exosomes regulate the process of polarization, Ye et al. reported that exosomes delivered DUSP2 and DUSP3 to macrophages and inhibited the activation of the P38 MAPK signaling pathway, thus promoting M2 polarization of macrophages [58]. However, more specific mechanisms remain to be further studied.

Secondly, exosomes could improve the histological characteristics of the tendon. Studies have reported that exosomes could promote the migration, proliferation and fibrosis of tenocytes, thus promoting tendon healing [56,67,123,124,125]. This process is closely related to TGF-β1, VEGFA and several kinds of microRNA. Exosomes could promote the expression of collagen fibers, make the fibers more orderly and increase the ratio of type I/III collagen [59,72]. The most abundant collagen in a tendon is type I collagen, which has a stiff structure and good strength [126]. However, after tendon injury, the expression of both type I collagen and type III collagen increases, and type III collagen is mainly present in scars [127,128]. Exosomes promote tendon healing while reducing scar formation [59,72]. Therefore, improving the histological characteristics of tendons is also one of the mechanisms of exosomes promoting tendon healing.

In addition, exosomes promote the expression of tenogenesis factors and cytokines, reconstructing the ECM. Exosomes can promote the expression of tenogenesis genes such as TNMD, TNC, Scx, DCN and MKX, and ECM genes such as BGN and ACAN, while inhibiting the expression of metalloproteinase genes. It is the expression changes of these genes that increase the synthesis of matrix and fibers, while reducing their degradation, which contributes to the formation of ECM and provides a favorable environment for the proliferation and migration of tendon cells, thus promoting tendon healing [50,55,61,71,73].

### 4.6. Mechanism of Exosomes Promoting Tendon–Bone Healing

The mechanism of exosomes promoting tendon–bone healing can be summarized in the following four ways: 1. Inhibiting inflammatory reaction and regulating macrophage polarization. 2. Promoting the expression of some cytokines, thus promoting the reconstruction of cell phenotype gradients at the tendon–bone interface. 3. Promoting the expression of bone metabolism factors, thus promoting osteogenesis and inhibiting osteolysis. 4. Promoting angiogenesis.

Similar to tendon healing, exosomes promoting M2 polarization of macrophages and inhibiting inflammatory reactions also play an important role in tendon–bone healing. Seven studies of tendon–bone healing reported this effect [81,84,86,87,88,89,90], which is one of the mechanisms of exosomes promoting tendon–bone healing.

Exosomes promote the expression of some cytokines, thus promoting the reconstruction of cell phenotype gradients at the tendon–bone interface. As we know, the tendon–bone interface is a gradient structure that can be divided into four layers [20]. It is the gradient structure that can disperse the stress and make the tendon–bone interface have some mechanical strength [22]. Exosomes promoted the expression of Scx and SOX-9 [77,86,87], which were reported to regulate the differentiation of progenitor cells and promote the establishment of cell phenotype gradients of the tendon–bone interface [22,129]. Multiple studies have reported that the hedgehog proteins and the hedgehog signaling pathway played a significant role in the reconstruction of the tendon–bone interface gradient [130,131,132]. However, no studies have reported whether there is a regulatory relationship between exosomes and hedgehog proteins. For future research, we suggest changing the expression level of exosomes while detecting the expression level of hedgehog proteins to determine whether there is a regulatory relationship between the two. If the regulatory relationship exists, it is possible to upregulate the expression of exosomes while inhibiting the expression of hedgehog proteins to determine whether exosomes promote tendon healing through hedgehog proteins. Exosomes also promoted the expression of collagen fibers and ECM components [84,86]. For example, COL II is related to chondrogenesis, and Smad and other proteins regulate cell transcription through multiple signaling pathways, thus promoting the formation of ECM components and cell phenotype gradient at the tendon–bone interface [133,134].

Additionally, exosomes promote the expression of bone metabolism factors, thus promoting osteogenesis and inhibiting osteolysis. A previous study reported that osteolysis or bone loss decreased the stiffness and strength of the tendon–bone interface, which was not conducive to tendon–bone healing [135]. Therefore, promoting osteogenesis and inhibiting osteolysis are significant in tendon–bone healing. Several studies have reported exosomes promoted tendon–bone healing by promoting osteogenesis and inhibiting osteolysis [76,80,86,87]. Wang et al. reported that exosomes secreted by Scx-overexpressing BM-MSCs targeted OCSTAMP and CXCL12 via miR-6924-5p, which inhibited osteolysis and thus promoted tendon–bone healing. The expression of osteoclast markers such as ACP5, CALCR, NFATc1 and ITGB3 was also suppressed [80]. Han et al. reported that exosomes with BMP-2 promoted the expression of RUNX2 and Smad, improved the strength of the tendon–bone interface, and promoted tendon–bone healing through the Smad/RUNX2 pathway [86]. Ren et al. reported that purified exosome products promoted the proliferation, migration and fusion of tenocytes and osteoblasts in vitro [87]. Fu et al. reported that exosomes promoted the expression of RUNX2, SOX-9, and TNC, which regulated osteogenesis, chondrogenesis and tenogenesis [76].

Finally, angiogenesis is also one of the significant mechanisms of exosomes promoting tendon–bone healing. One of the difficulties in tendon–bone healing is that there are few vessels at the tendon–bone interface [24]. Blood vessels provide adequate oxygen and nutrients and remove metabolic wastes effectively, which may contribute to the formation of cell phenotype gradients [136]. Therefore, angiogenesis is crucial in the process of tendon–bone healing. However, few studies have reported that exosomes promote angiogenesis and thus promote tendon–bone healing. Huang et al. reported that animals treated with exosomes had more neovascularization at the injured site [88]. In their in vitro experiment, they found that exosomes activated the Hippo signaling pathway through VGEF, which promoted the proliferation and migration of HUVECs [88]. In the future, the role of angiogenesis in exosomes promoting tendon–bone healing needs to be further studied. In current studies, it is rare to see study on the role of angiogenesis in tendon–bone healing in animal models. Future researchers can consider making breakthroughs in this area.

### 4.7. Inconsistency and Possible Explanations

#### 4.7.1. Inconsistency between Different Studies

It was mentioned before that the roles of exosomes in tendon adhesion and peritendinous fibrosis were controversial in several studies and the expression of α-SMA was up-regulated in some studies while down-regulated in some other studies [9,49,56,63,68,83]. Previous studies have reported that TGF-β1 promoted the expression of α-SMA and the proliferation of fibroblasts, and thus enhanced the formation of ECM and tendon adhesion [125,137]. Yao et al. reported that exosomes inhibited the roles of TGF-β1 and thus inhibited the proliferation of fibroblasts via miR-21a-3p [9]. Similarly, Li et al. also reported that exosomes inhibited the promoting effects of TGF-β on the proliferation of fibroblasts and the expression of α-SMA [63]. Conversely, Cui et al. reported that miR-21-5p in exosomes from macrophages targeted and inhibited Smad7, resulting in the activation of the TGF-β1 signaling pathway, and thus the migration and proliferation of tenocytes and fibroblasts were promoted [56]. Yu et al. reported that circRNA-Ep400 was expressed in exosomes from M2 macrophages, and the exosomes promoted the expression of TGF-β1 and peritendinous fibrosis via the miR-15b-5p/FGF-1/7/9 pathway [68]. Wu et al. also reported that miR-21-5p was abundant in IONP-exosomes, which targeted Smad7 and activated the TGF-β1/Smad pathway, thus promoting the formation of ECM and tissue fibrosis [83]. Zhang et al. also reported that the expression of α-SMA was down-regulated but did not explore the mechanism [49].

#### 4.7.2. Reasonable Explanations

We could draw a preliminary conclusion that exosomes from different sources contain different non-coding RNAs, which may act on different signaling pathways, and thus promote or inhibit the function of TGF-β1. Therefore, different exosomes may have different effects on peritendinous fibrosis and the expression of α-SMA. It is evident that miR-21-5p and circRNA-Ep400 up-regulate TGF-β1 activity and promote peritendinous fibrosis, while miR-21a-3p negatively regulates TGF-β1 activity and inhibits peritendinous fibrosis. We have mentioned before that there were few differences between the biomechanical outcomes of groups in these studies [9,56,63]. We speculate that exosomes promoted tendon fibrosis and adhesion, which maintained the mechanical strength of the tendon. A study reported that TGF-β1 also promoted scar formation, and inhibition of expression of α-SMA may be related to the inhibition of scar formation [138]. That exosomes inhibit scar formation seems to be a reasonable explanation. So, different exosomes can mediate TGF-β1 to play different roles, which may be related to the sources, concentration, and acting time of exosomes. However, the exact mechanism remains to be explored.

#### 4.7.3. Future Research Directions

It is clear that exosomes have both advantages and disadvantages for tendon and tendon–bone healing. In some cases, exosomes promote the expression of collagen fibers and fibrocartilage while inhibiting the adhesion of the tendon and surrounding tissues, thus promoting tendon and tendon–bone healing. However, in other cases, exosomes may excessively promote fibrosis, leading to the adhesion of a tendon and scar formation. This may be related to the source, concentration, acting time and the type of non-coding RNA of exosomes, which is a significant research direction in the future. With the continuous development of technology, the application of computational simulation and computer technology in medical science research is becoming increasingly widespread. Computer technology has lower costs and faster results, and the full application of computer technology can further verify the reliability of preclinical and clinical studies. Therefore, computer technology has broad application prospects and needs to be paid attention to in future research.

### 4.8. Limitations

Our systematic review also has some limitations. Firstly, the quality of studies included limited the persuasiveness of this review. Few studies reported the methodology of sequence generation, allocation concealment, random housing or blinding methods. There were some unclear risks in these studies according to the SYRCLE risk of bias assessment tool [101]. Therefore, standardization of outcome reporting should be established, and more detailed documentation of the methodology should be demonstrated in future studies. Secondly, researchers still lack a certain understanding of exosomes as a therapeutic drug. The most suitable source, isolation methods, concentration and administration frequency of exosomes are still unknown. The results of the included studies are not convincing enough. Furthermore, the monitoring of biological activity changes and the pharmacokinetics of exosomes were lacking in the studies. Thirdly, the animal models of studies also limit the power of this review. The animals used to establish the tendon injury models were relatively small and young, which cannot simulate the complete process of tendon healing in humans. Additionally, the biomechanical test indexes need to be improved. Finally, the heterogeneity of the reporting form of outcomes precluded a more rigorous analysis of the studies. Due to the lack of uniform reports of the outcomes, especially quantitative reports, this systematic review cannot use meta-analysis to further analyze the results. In the future, the reports on the methodology and outcomes need to be standardized. In this way, more conclusions could be drawn from the studies.

## 5. Conclusions

We systematically assessed the existing preclinical animal studies on tendon and tendon–bone healing, and demonstrated that it is promising to use exosomes to promote tendon healing and tendon–bone healing. These findings provide basic support for the clinical translation of exosomes as a tendon and tendon–bone healing therapy. For future work, researchers can focus on the following aspects: 1. Specific mechanisms by which pre-conditioning of MSCs or exosomes enhances the therapeutic effect of exosomes. 2. The most suitable method, concentration, and frequency of the administration of exosomes. 3. Researchers can use larger animal models to simulate human tendon injuries more realistically. 4. The detailed molecular biological mechanisms of exosomes promoting tendon healing and tendon–bone healing. 5. Using computer technology to simulate clinical and preclinical studies. However, the unclear-to-low risk of bias highlights the significance of the standardization of outcome reporting. Further preclinical studies are still needed to produce the most suitable exosomes which promote tendon and tendon–bone healing while inhibiting adhesion and scar formation for future clinical studies.

## Figures and Tables

**Figure 1 jfb-14-00299-f001:**
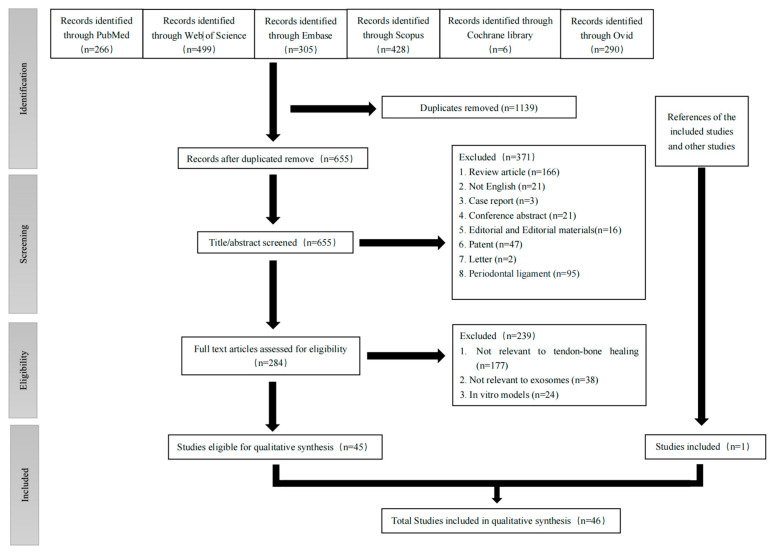
PRISMA (Preferred Reporting Items for Systematic Reviews and Meta-Analyses) flow chart.

**Table 1 jfb-14-00299-t001:** Summary of existing treatment options for acute tendon injury and chronic tendon injury.

Injury Types	Treatment Options
Acute tendon injury	Existing treatment options for acute tendon injury are mostly surgical treatment. Surgical treatment includes open, minimally invasive, and percutaneous repair techniques [14]. Currently, biological agents such as platelet-rich plasma are also used in surgical treatment to promote postoperative recovery of tendons [15].
Chronic tendon injury	Existing treatment options for chronic tendon injury include activity modification, relative rest, drug treatment and rehabilitative exercise [16]. Traditional drugs used to treat chronic tendon injuries include non-steroidal anti-inflammatory drugs (NSAIDs), corticosteroids and topical nitroglycerin [17]. New methods such as ultrasound-guided debridement and platelet-rich plasma are also used to address chronic tendon injuries [18,19].

**Table 2 jfb-14-00299-t002:** Summary of existing cellular therapy options for tendon injury.

Cellular Therapy Options	Overview of the Options
Mesenchymal stem cells (MSCs)	MSCs are of great significance for tendon regeneration. MSCs have a wide range of sources and can be obtained from tissues such as fat, umbilical cord, and bone marrow. MSCs promote tendon regeneration by secreting cytokines, regulating inflammation, and differentiating into tendon cells [91]. Currently, MSCs used for treatment include Adipose-derived mesenchymal stem cells (ASCs), Bone marrow mesenchymal stem cells (BMSCs), and tendon stem cells (TSCs). The conventional vacuum fraction (SVF) also contains a certain amount of MSCs, which are currently being studied [92].
Growth factors	Growth factors can promote cell proliferation and differentiation, and stimulate the synthesis of extracellular matrix (ECM), which are beneficial for tendon regeneration. Common growth factors related to tendon regeneration include insulin-like growth factor 1 (IGF-1), vascular endothelial growth factor (VEGF), bone morphogenetic proteins (BMPs) and so on [93].
Platelet-rich plasma (PRP)	PRP contains various growth factors that can promote tendon regeneration by promoting cell proliferation and angiogenesis [94]. PRP has certain clinical application potential.
Extracellular vesicles (EVs)	Currently, the EVs mainly used for studying the promotion of tissue regeneration are exosomes. EVs contain a variety of proteins, lipids and nucleic acids such as microRNA (miRNA), mRNA and DNA, which play an important role in cell-to-cell communication [40].
Differentiated cells	Differentiated cells include Tenocytes and fibroblasts. Differentiated cells have strong proliferation and differentiation abilities, and they will not lead to teratoma, which is their major advantage [95].
Pluripotent stem cells (PSCs)	PSCs include embryonic stem cells (ESCs) and induced pluripotent stem cells (iPSCs), they have a strong ability for proliferation and differentiation. PSCs differentiate to functional tendon cells to promote tendon regeneration [96].

**Table 3 jfb-14-00299-t003:** Summary of characteristics of animal models used (tendon healing).

Author	Year	Animal	Sample Size	Gender	Animal Model
Chamberlain et al. [64]	2019	Nude mice	27	Male	The SDF was removed from the Achilles tendon, and the Achilles tendon was completely transected at the midpoint, followed by repair.
Chen et al. [61]	2021	NZ rabbits	NR	NR	The Achilles tendon was transected, followed by repair with the Modified Kessler four-core technique.
Cui et al. [56]	2019	C57BL/6J mice	NR	Male	Complete transection and repair of the FDL tendon were conducted in the right hind paw
Davies et al. [54]	2022	C57BL/6J mice	12	NR	The supraspinatus and infraspinatus tendons were transected and the suprascapular nerve was identified and resected.
Fu et al. [76]	2021	SD rats	96	NR	The rats’ supraspinatus tendon was cut off on Bilateral shoulders and cartilage was worn down. Repair was conducted 4 weeks later.
Gao et al. [51]	2022	SD rats	9	Male	Carrageenan was injected into quadriceps tendon under ultrasound guidance.
Gissi et al. [62]	2020	Lewis rats	16	Male	A bilateral Achilles tendon defect 2 mm in diameter was conducted.
Han et al. [69]	2022	SD rats	18	Male	The superficial Achilles tendon was removed, and transection was made in the middle of the deep Achilles tendon, followed by repair.
Hayashi et al. [48]	2022	C57BL/6J mice	39	Male	A transverse incision was made at the midpoint of the Achilles tendon with the scissors.
Jenner et al. [70]	2023	Sheep	12	NR	A full-thickness defect was created in the center of the infraspinatus tendon at the enthesis.
Li et al. [63]	2020	SD rats	33	Male	The thin strand of the Achilles tendon was resected, and the thick strand was transected in the middle, followed by repair of the tendon.
Li et al. [71]	2022	SD rats	64	Male	One-third of the central part of the patellar tendon was removed.
Liu et al. [77]	2021	SD rats	63	Male	One-third of the central part of the patellar tendon was removed.
Liu et al. [53]	2021	Rats	20	NR	Type I collagenase solution was injected into the right hind leg of the rats every 2 days for 2weeks.
Shen et al. [50]	2019	NGL mice	32	Both sexes	At the midpoint between the musculotendinous junction of the Achilles tendon and the calcaneal insertion, the Achilles tendon was transected.
Shi et al. [52]	2020	Dogs	48	NR	The tendon was transected in the middle of a 30mm length centered at the proximal interphalangeal joint level.
Song et al. [67]	2022	SD rats	72	Male	One-third of the central part of the patellar tendon was removed
Tao et al. [8]	2021	SD rats	96	Male	One third of the central part of the left knee patellar tendon was removed.
Wang et al. [60]	2022	NZ rabbits	112	Male	The bursal-side defects on the supraspinatus tendon were made to establish the 50% PTRCT model.
Wang et al. [65]	2019	SD rats	42	Female	The supraspinatus and infraspinatus tendons were cut close to the greater tuberosity of the humerus.
Wang et al. [66]	2019	SD rats	18	Male	Type I collagenase solution was injected into Achilles tendons.
Wang et al. [79]	2021	C57BL/6J mice	72	Male	The mice were trained on a treadmill for 1 week.
Wellings et al. [59]	2021	NZ rabbits	45	Female	The Achilles tendon bundle was isolated and transected approximately 1.5 cm proximal to the calcaneal tubercle. Then, tendon was repaired with Kessler core suture technique.
Xu et al. [72]	2022	SD rats	36	Male	Type I collagenase solution was injected into the Achilles tendon.
Xu et al. [74]	2023	SD rats	100	Male	Achilles tendon was completely full-thickness ruptured, followed by repair.
Yao et al. [9]	2020	SD rats	60	Male	The superficial Achilles tendon was removed, and transection was made in the middle of the deep Achilles tendon, followed by repair.
Yao et al. [55]	2021	SD rats	180	Male	A rectangular full-thick-ness defect was introduced to the left Achilles tendon.
Ye et al. [58]	2023	SD rats	30	Male	Carrageenan solution was injected around the right quadriceps tendon.
Yu et al. [68]	2021	C57BL/6J mice	24	Male	The FDL tendons in the mouse right hind paw were transected and repaired
Yu et al. [73]	2020	SD rats	52	Male	One-third of the central part of the patellar tendon was removed.
Zhang et al. [49]	2020	SD rats	54	Male	One third of the central part of the Achilles tendon was removed.
Zhang et al. [57]	2022	NZ rabbits	108	Male	The supraspinatus tendon was cut from the greater tubercle, and the torn tendon was left unrepaired for 12 weeks.
Zhang et al. [75]	2022	SD rats	36	Male	The supraspinatus tendon was completely cut from the greater tubercle to create a full-thickness injury, which was left unrepaired for 12 weeks.
Zhao et al. [78]	2022	SD rats	NR	NR	The rat tendons were cut off for one week.
Zhu et al. [47]	2022	SD rats	20	Female	Carrageenan solution was injected around the quadriceps tendon.

SD rats, Sprague Dawley rats; C57BL/6J, C57 black 6 Jackson Laboratory; NR, not reported; NGL mice, NF-κB-GFP-luciferase transgenic reporter mice; FDL, flexor digitorum longus; NZ, New Zealand; SDF, superficial digital flexor.

**Table 4 jfb-14-00299-t004:** Summary of characteristics of animal models used (tendon–bone healing).

Author	Year	Animal	Sample Size	Gender	Animal Model
Han et al. [86]	2022	Rabbits	30	Male	The supraspinatus tendon was cut, and a 0.5 cm × 0.5 cm tendon tissue was removed.
Huang et al. [88]	2020	SD rats	54	Male	The supraspinatus insertion was resected, and some of the supraspinatus was cut off.
Li et al. [89]	2022	Wistar rats	90	Male	Unilateral ACL resection (at the right side), and bone tunnels were created.
Ren et al. [87]	2021	SD rats	36	NR	The supraspinatus tendon was transected at its insertion site on the greater tuberosity.
Shi et al. [84]	2020	C57BL/6J mice	90	Male	The Achilles tendon was cut off, and the cartilage layer at the insertion was removed.
Wang et al. [80]	2021	C57BL/6J mice	NR	NR	The Achilles tendon was released from the calcaneal tuberosity to the calf muscle and a 3 mm midline knee incision was made. A tunnel was drilled in the proximal tibial metaphysis to the long axis of the tibia.
Wang et al. [90]	2020	Rabbits	35	Male	The supraspinatus tendon was detached at the insertion on the greater tuberosity of the humerus.
Wu et al. [83]	2022	SD rats	108	Male	Unilateral ACL resection, and bone tunnels were created.
Wu et al. [85]	2021	C57BL/6J mice	120	Male	The supraspinatus tendon was transected at its insertion site on the greater tuberosity, and the remaining tendon and fibrocartilage layer of the footprint were gently abraded.
Xu et al. [81]	2022	SD rats	90	Male	The intra-articular ACL was sectioned, and bone tunnels were created.
Zhang et al. [82]	2022	SD rats	78	Male	Unilateral ACL resection (at the right side), and bone tunnels were created.

SD rats, Sprague Dawley rats; C57BL/6J, C57 black 6 Jackson Laboratory; NR, not reported; ACL, anterior cruciate ligament.

**Table 5 jfb-14-00299-t005:** Summary of key therapeutic outcomes (tendon healing).

Author	Year	In Vivo Outcomes	In Vitro Outcomes
Chamberlain et al. [64]	2019	EEM treatment significantly increased ultimate stress and Young’s modulus, and EVs treatment was not as effective as EEM treatment in improving biomechanical properties. The treatment of macrophages, EEMs, and EVs all significantly down-regulated type I collagen expression and decreased the M1/M2 macrophage ratio.	NR
Chen et al. [61]	2021	The biomechanical properties were enhanced with the treatment of rabbit ASC exosomes. Rabbit ASC exosomes reduced the inflammatory hardening, promoted the expression of D CN, COLI, TNMD and BGN, inhibited the expression of type III collagen, and made the collagen fibers more orderly.	HU-MSC exosomes promoted the proliferation and migration of tenocytes in a dose-dependent manner.
Cui et al. [56]	2019	There were no significant differences in the biomechanical properties of the tendons between the 2 groups. Mouse BMDM exosomes induced fibrosis of injured tendon, upregulated the expression of COL I, COL III, α-SMA and TGF-β1, and caused adhesion between tendon and surrounding tissues.	Mouse BMDM exosomes promoted proliferation and migration of fibroblasts and tenocytes and improved their fibrotic activity, and up-regulated the expression levels of COL I, COL III, α-SMA and TGF-β1 via miR-21-5p which directly targets Smad7.
Davies et al. [54]	2022	Fat infiltration was significantly reduced in the exosomes treated group.	Mouse FAP exosomes promoted cell proliferation, migration and differentiation.
Fu et al. [76]	2021	Hydrogel group and EHC group had less inflammatory reaction, while EHC group had more regular fiber arrangement and the best biomechanical properties. EHC significantly up-regulated the expression of RUNX-2, SOX-9 and TNC. Hydrogel also up-regulated their expression, but not as obvious as EHC.	Human ASC exosomes promoted the proliferation and differentiation of TSCs in a dose-dependent manner.
Gao et al. [51]	2022	Human IP-MSC exosomes down-regulate the expression of CGRP, iNOS and other inflammatory cytokines, and can significantly relieve the pain caused by tendon lesions. The group treated with exosomes showed a higher histological score.	Human IP-MSC exosomes inhibited mast cell activation and down-regulated inflammatory cytokines by regulating HIF-1 signaling pathway.
Gissi et al. [62]	2020	High concentration of EVs significantly promoted the expression of type I collagen fibers and inhibited the expression of type III collagen fibers, and the group had higher histological scores and a more obvious promotion effect on tendon healing.	High concentration of EVs promoted the expression of type I collagen and the proliferation of tenocytes, and both high and low concentration of EVs promoted the migration of tenocytes. MMP 14 was present in EVs.
Han et al. [69]	2022	HU-MSC exosomes made Achilles tendon arrange more orderly, cells proliferated well, and effectively alleviated tendon injury.	HU-MSC exosomes promoted the proliferation and migration of tenocytes and activate RhoA in a dose-dependent manner. HU-MSC exosomes reduced tendon injury via miR-27b-3p-mediated suppression of ARHGAP5, resulting in RhoA activation.
Hayashi et al. [48]	2022	P5 EVs promoted the formation of collagen fibers and the growth of fibrous tissue was good. The P5 EVs treatment group had better histological scores, good tendon healing, and no significant adhesions.	P5 EVs promoted the proliferation and migration of tenocytes more significantly.
Jenner et al. [70]	2023	Sheep treated with HU-MSC exosomes had improved orientation of collagen fibers and less osteophyte formation at the injury site. The fibrocartilaginous transition zone was formed, inflammation at the lesion site was alleviated and fibrotic adhesions were significantly reduced	HU-MSC exosomes inhibited the proliferation of CD3/CD28 stimulated T-cells.
Li et al. [63]	2020	Tendon maximum tensile strength remained the same in all three groups. Both EVs and HCPT-EVs reduced the degree of adhesion between tendon and surrounding tissues, while the group treated with HCPT-EVs had the best histological score.	HCPT-EVs up-regulated the expression of GRP78, CHOP and Bax, and down-regulated the expression of Bcl-2, COL III, α-SMA, which might activate ERS pathway to inhibit adhesion. HCPT-EVs reduced the activity of fibroblasts and inhibited their proliferation more effectively.
Li et al. [71]	2022	TSC treated with EUG-BM-MSC-EVs significantly improved tendon fiber arrangement, promoted the expression of type I collagen and type III collagen. The expressions of PCNA, TNMD, bFGF and SCXA were higher than those in other groups.	EUG-BM-MSC-EVs promoted the proliferation and migration of TSC and increased the expression of PCNA. Additionally, EUG-BM-MSC-EVs reversed the down-regulation of Col I, TNC, TNMD and SCXA expression induced by H_2_O_2_, and decreased the apoptosis rate, PARP1 expression and ROS content.
Liu et al. [77]	2021	Rat ASC exosomes significantly increased ultimate load, stiffness, and Young’s modulus, reduced inflammatory reaction, made collagen fibers more organized and tightly packed, and upregulated the expressions of TNMD, COL I, SCXA, and M2 macrophage markers.	Rat ASC exosomes promoted TSCs proliferation and migration and upregulated the expression of TNMD, COL I and SCXA by activating Smad2/3 and Smad1/5/9 signaling pathways.
Liu et al. [53]	2021	Rat TSC exosomes promoted the expression of type I collagen, inhibited the expression of type III collagen, and inhibited inflammatory reaction. The treatment effect of EXO/MBA group was better than that of EXO group.	EXO/MBA promoted the proliferation and differentiation of tenocytes, promoted the expression of collagen fibers, and effectively inhibited the degradation of extracellular matrix and inflammatory reaction. Exosomes also down-regulated the expression of MMP-3 and MMP-13.
Shen et al. [50]	2019	Mouse ASC IEVs decreased the activity of NF-κB, up-regulated the expression of COL I, COL II, COL III, SOX-9, and down-regulated the expression of MMP-1.	Both IEVs and EVs effectively inhibited the inflammatory response, but the effect of IEVs was stronger.
Shi et al. [52]	2020	TEPEP patch significantly improved the load-failure strength and tensile stiffness of the tendon, and increased the expression of collagen fibers. In the group treated with TEPEP patch, there were a large number of fibroblasts migrated to the injury site, and the number of cells was the largest among the three groups.	TEPEP patch up-regulated the expression levels of COL III, MMP-2, MMP-3 and MMP-14 in tenocytes and inhibited inflammatory response.
Song et al. [67]	2022	The injection of rat TSC exosomes significantly promoted tendon healing and the recovery of biomechanical properties which included the ultimate stress and Young’s modulus of injured tendon. Histologically, the group treated with exosomes had the best histological score with more regular fiber alignment.	Rat TSC exosomes promoted the proliferation and migration of tenocytes and the expression of type I collagen, SCX, COL I and DCN through miR-144-3P. High concentration of exosomes could protect tenocytes from oxidative stress and serum deprivation in vitro.
Tao et al. [8]	2021	Different EVs have different effects on the biomechanical properties of the interface and the formation of various tissues. H19-OL-EVs demonstrated the best effect on promoting the formation and arrangement of matrix and collagen.	Three kinds of pretreated EVs had more significant effects on the proliferation, migration and differentiation of TSPCs and the activation of YAP. H19 regulated YAP phosphorylation and translocation through H19-pp1-YAP interactions, thereby promoting proliferation, migration, and expression of tendon-related genes.
Wang et al. [60]	2022	Human ASC exosomes significantly improves ultimate failure load, stiffness, and ultimate tensile strength. Histologically, the group treated with exosomes contained more well-aligned collagen fibers, had a better histological score, and had a higher intensity and amount of type I collagen than the other groups.	NR
Wang et al. [65]	2019	The injection of exosomes can prevent the decrease of biomechanical properties. Exosomes reduced the fatty infiltration, inflammatory reaction and apoptosis rate of the tissues. The degree of vascularization in exosomes group was lower than that in saline group.	NR
Wang et al. [66]	2019	Rat TSC exosomes enhanced the maximum loading and ultimate stress of injured tendon. Histologically, the group treated with exosomes had more homogeneous collagen arrangement, better histological score, lower MMP-3 and higher TIMP-3 and COL I expression.	Rat TSC exosomes reversed the inflammatory reaction induced by IL-b, inhibited the expression of MMP-3, and promoted the expression of TIMP-3 and COL I.
Wang et al. [79]	2021	Human ASC exosomes significantly increased the maximum failure load of the tendon. Human ASC exosomes up-regulated the expression of type I collagen and the ratio of collagen I/III, down-regulated the expression of type III collagen, and inhibited the formation of adhesion and contracture. Human ASC exosomes promoted M2 polarization of macrophages and down-regulated the expression of MMP-3, MMP-13 and SOX-9.	NR
Wellings et al. [59]	2021	The failure load and ultimate tensile stress of the three groups were similar. PEP-treated tendons contained more well-aligned collagen fibers, lower adhesion grades, and a ratio of type I and type III collagen more similar to normal tendon.	NR
Xu et al. [72]	2022	Rats treated with exosomes showed less inflammatory reaction, more mature collagen fibers, less angiogenesis, higher expression of type I collagen, lower expression of type III collagen, higher biomechanical properties and lower percentage of lesions than rats in other groups.	MiR-29a, miR-21-5p and miR-148a-3p are highly expressed in Exosomes
Xu et al. [74]	2023	In the rat Achilles tendon injury model, both EV_N_ and EV_B_ significantly improved the biomechanical properties of the tendon, but EV_B_ restored the intrinsic failure pattern. EV_B_ promoted capillary formation and vessel maturation during tendon regeneration, promoted M2 polarization of macrophages, up-regulated the expression of SCX and TNMD, and reduced scar formation and detrimental morphological changes in the Achilles tendon.	EV_N_-educated macrophages promoted endothelial cell migration and angiogenesis in vitro. EV_B_-educated macrophages upregulated VEGF expression and promoted M2 polarization of macrophages. Levels of HSA-miR-125a-5p, HSA-miR-199b-3p, and miR-92b-5p were elevated in EV_B_, the first two of which were reported to regulate macrophage polarization and promote angiogenesis.
Yao et al. [9]	2020	There was no significant difference in maximum tensile strength among the three groups. The group treated with HU-MSC exosomes had the best histological score, the least collagen deposition, and the adhesion of tendon and surrounding tissue was effectively relieved.	HU-MSC exosomes manipulated p65 activity by delivering low-abundance miR-21a-3p, thus the expression of COL III and α-SMA was down-regulated, and the proliferation of fibroblasts was inhibited. Ultimately, tendon adhesion was inhibited.
Yao et al. [55]	2021	HU-MSC exosomes up-regulated the expression of tendon markers such as COL I, TNMD and SCXA, down-regulated the expression of COL III, and promoted the deposition of extracellular matrix in tendon, thus promoting tendon healing. Additionally, HU-MSC exosomes significantly improve the biomechanical properties of tendons through miR-29a-3p.	HU-MSC exosomes regulated PTEN/mTOR/TGF-β1 pathway through miR-29a-3p to up-regulate the expression of tendon-related genes and promote TSC differentiation into tendon.
Ye et al. [58]	2023	Large EVs and Small EVs had similar functions, including relieving the pain of rat tendon, inhibiting the inflammatory reaction and improving the histological score of diseased tendons.	Large EVs promoted M2 polarization of macrophages in a dose-dependent manner and inhibited inflammatory responses. Large EVs delivered encapsulated DUSP2 and DUSP3 to macrophages, inhibited the activation of P38 MAPK signaling pathway, and thus promoted M2 polarization of macrophages.
Yu et al. [68]	2021	Ultimate stress, Young’s modulus, and tensile strength were increased with the treatment of mouse BMDM exosomes, while this therapeutic effect was inhibited by the downregulation of CircRNA-EP400. Mouse BMDM exosomes promoted peritendinous fibrosis by CircRNA-EP400.	Mouse BMDM exosomes, especially mouse BMDM exosomes with high CircRNA-EP400 expression, promoted the proliferation and migration of fibroblasts and tenocytes through miR-15b-5p/FGF-1/7/9 pathway, and increased the expression levels of FGF-1, FGF-7, FGF-9, TGF-β1, TGF-I and α-SMA.
Yu et al. [73]	2020	Rat BM-MSC exosomes increased the deposition of type I collagen and the density and arrangement of cells in the injured area were more similar to those in normal tendon. The histological score was better, and the expression of TNMD, number of TSPCs and biomechanical properties were increased.	Rat BM-MSC Exosomes promoted the proliferation, migration and differentiation of TSPCs and the expression of TNMD, MKX and COL I genes.
Zhang et al. [49]	2020	Rat TSC exosomes promoted the formation of collagen fibrils, and their arrangement was more continuous and regular. Additionally, exosomes promoted M2 polarization of macrophages and inhibited inflammatory response.	Rat TSC exosomes significantly promoted tenocytes proliferation and migration in a dose-dependent manner through the activation of PI3K/AKT and MAPK/ERK1/2 signaling molecules. Rat TSC exosomes up-regulated the expression of COL I, COL III and TIMP-1, down-regulated the expression of MMP-9 and α-SMA.
Zhang et al. [57]	2022	The REPA group contained more well-aligned fibers and fibrocartilage, and always showed a lower fat infiltration than other groups	Rabbit BM-MSC exosomes with patch promoted the proliferation and migration of tenocytes.
Zhang et al. [75]	2022	Injection of exosomes and GC counteracted the negative effect of GC on the biomechanical properties of injured tendons. The rats injected with GC showed fatty infiltration and collagen degeneration, and their histological characteristics were significantly worse than those of rats in the control group and rats injected with GC and exosomes.	GC reduced the inflammatory reaction, but inhibited the proliferation and migration of tenocytes, down-regulated the expression of collagen, up-regulated the expression of MMP-2, MMP-9 and MMP-13, and promoted cell senescence and apoptosis. Exosomes further reduced the inflammatory reaction and counteracted the negative effects of GC. In addition, exosomes promoted cell proliferation, up-regulated collagen expression and type I/III ratio
Zhao et al. [78]	2022	The group treated with exosomes had less inflammatory reaction, better structure, lower caspase-3 expression and higher PCNA expression and activity of tendon cells. Exosomes decreased the levels of CK, LDH, MDA and oxidative stress.	IGFBP3 promoted the expression of CK, LDH, MDA and caspase-3, and inhibited the expression of PCNA. Exosomes inhibited IGFBP3 expression through MiR-19a, which promoted tenocytes proliferation and decreased apoptosis rate.
Zhu et al. [47]	2022	Human IP-MSC exosomes can significantly relieve chronic pain caused by tendinopathy. The group treated with exosomes has higher histological scores, and the inflammatory response and capillary proliferation were inhibited.	Human IP-MSC exosomes promoted the proliferation of tenocytes and the expression of anti-inflammatory cytokines, and down-regulated the expression of pro-inflammatory cytokines

IP-MSC, induced pluripotent mesenchymal stem cells; P5 EVs, extracellular vesicles from mesenchymal stem cell at passage 5; TSC, tendon stem cell; PI3K, Phosphatidylinositol3-kinase; AKT, protein kinase B; MAPK, mitogen-activated protein kinase; ERK, extracellular regulated protein kinases; COL I: Collagen type I; COL III: Collagen type III; TIMP, Tissue inhibitor of the metalloproteinase; MMP, Matrix metalloproteinases; α-SMA: Alpha-smooth muscle actin; ASC, Adipose stem cell; IEVs, IFN γ-primed Extracellular vesicles; NF-κB, Nuclear Factor-kappa B; COL II: Collagen type II; SOX-9, sex determining region Y box 9; EVs, extracellular vesicles; H19-OL-EVs, extracellular vesicles derived from tendon stem/progenitor cells co-overexpressed of H19 and hnRNP A2/B1; TSPCs, tendon stem/progenitor cells; YAP, yes-associated protein; CGRP, Calcitonin Gene-Related Product; iNOS, inducible Nitric Oxide Synthase; HIF-1, Hypoxia-inducible factor-1; TEPEP, Tisseel plus purified exosome product; EXO, Exosomes; EXO/MBA, exosomes modified by a nitric oxide nanomotor; FAP, Fibroadipogenic progenitor; HU-MSC, Human umbilical cord mesenchymal stem cell; TNMD: Tenomodulin; SCX, Scleraxis; TGF-β1: Transforming growth factor-β1; PTEN: Phosphatase and tensin homolog; mTOR, mammalian Target of Rapamycin; BMDM, bone-marrow-derived macrophage; Smad, Mothers Against Decapentaplegic Homolog; REPA, repaired with exosomes loaded patch augmentation; BM-MSC, bone morrow mesenchymal stem cells; DUSP, Dual Specificity Phosphatase; PEP, purified exosome product; DCN, Decorin; BGN, biglycan; HCPT-exosomes, exosomes derived from human umbilical cord mesenchymal stem cell treated with hydroxycamptothecin; GRP, glucose regulated protein; ERS: Endoplasmic reticulum stress; EEM, exosome-educated macrophages; IL, interleukin; FGF, fibroblast growth factor; ARHGAP, Rho GTPase activating protein; EUG-BM-MSC-EVs, eugenol treated-bone morrow mesenchymal stem cells extracellular vesicles; PCNA, proliferating cell nuclear antigen; TNC, Tenascin C; PARP, poly ADP-ribose polymerase; ROS, reactive oxygen species; MKX, Mohawk; EV_B_, bioactive glasses-elicited mesenchymal stem cell extracellular vesicle; EV_N_, native mesenchymal stem cell extracellular vesicle; HSA, Human Serum Albumin; GC, Glucocorticoid; EHC, Exosomes-hydrogel complex; RUNX, Runt-related Transcription Factor; CK, Creatine Kinase; LDH, lactate dehydrogenase; MDA, Malondialdehyde; IGFBP, Insulin-like Growth Factor Binding Protein.

**Table 6 jfb-14-00299-t006:** Summary of key therapeutic outcomes (tendon–bone healing).

Author	Year	In Vivo Outcomes	In Vitro Outcomes
Han et al. [86]	2022	BMD, TMD, BV/TV were the highest in the BMP-2-EXO group where ultimate load strength and stiffness were also significantly increased. BMP-2-Exosomes inhibited the inflammatory reaction and promoted the formation of fibrocartilage. Chondrocytes arranged in order and the interface was similar to the natural tendon–bone interface. BMP-2-Exosomes up-regulated the expression of Smad4, Smad5, RUNX2, Aggrecan, COL II, SOX-9 and TIMP-1.	NR
Huang et al. [88]	2020	Rat BM-MSC exosomes up-regulated the expression of COL I, COL II and proteoglycan, and significantly increased the maximum breaking load and stiffness of the tendon. The group treated with exosomes had more neovascularization and better growth of tendon–bone interface.	Rat BM-MSC exosomes activated the Hippo signaling pathway through VGEF, which promoted the proliferation and migration of HUVECs. Exosomes inhibited the M1 polarization of macrophages and inhibited the release of pro-inflammatory cytokines.
Li et al. [89]	2022	The group treated by exosomes secreted by miR-23a-3p-overexpressing BM-MSCs had the smallest mean bone tunnel area, the largest BV/TV, the decreased relative width of the interface, and the largest amount of fibrocartilage. These exosomes promoted the proliferation of chondrocytes and the expression of COL II, and the maximal failure load and stiffness of the tendon were significantly increased.	Rat BM-MSC Exosomes inhibited inflammation by promoting M2 polarization of macrophages via miR-23a-3p.
Ren et al. [87]	2021	PEP promoted fibrocartilage and angiogenesis, increased stiffness and maximum tensile load, and inhibited inflammation. The interface was similar to the natural tendon–bone interface. PEP upregulated the expression of COL I, COL III, SCX, TNMD, TNC, DCN and IGF.	PEP significantly promoted the proliferation and migration of tenocytes and osteoblasts, and the fusion time was the shortest. PEP upregulated the expression of COL I, COL III, TNC, DCN, SCX, Spp1, EGR and PPARG.
Shi et al. [84]	2020	Rat BM-MSC exosomes promoted the formation of transitional structures at the tendon–bone interface and enhanced the biomechanical properties of the tendon–bone interface. The chondrocytes increased and the collagen fibers arranged in order. Exosomes up-regulated the expression of COL II, aggrecan, TGF-β 3, IGF-1 and IGF-2.	Rat BM-MSC exosomes promoted M2 polarization of macrophages and inhibited inflammatory response.
Wang et al. [80]	2021	Both miR-6924-5p and exosomes secreted by SCX-overexpressing BM-MSCs significantly inhibited osteolysis, prevented osteolysis and improved the biomechanical strength of tendon–bone interface.	Exosomes secreted by SCX-overexpressing BM-MSCs targeted OCSTAMP and CXCL12 via miR-6924-5p, inhibiting osteoclastogenesis.
Wang et al. [90]	2020	Human ASC exosomes significantly increased ultimate failure load, stiffness and stress. Compared with the group treated with saline, the group treated with exosomes had less fatty infiltration, milder inflammatory reaction, more fibrocartilage and type I collagen fibers, and the tendon–bone interface was more continuous and uniform.	NR
Wu et al. [83]	2022	The groups treated with exosomes demonstrated smaller mean bone tunnel area, larger BV/TV, and better graft to bone fusion. Exosomes reduced the width of the interface, promoted the formation of fibrocartilage, up-regulated the expression of α-SMA and OCN, and enhanced the biomechanical strength of the tendon–bone interface.	IONP-exosomes promoted NIH3T3 fibroblasts proliferation and migration by down-regulating Smad7 via miR-21-5p, which up-regulated the expression of Col I, Col III and α-SMA.
Wu et al. [85]	2021	The ultimate failure load and stiffness of the LIPUS-BM-MSC-Exosomes group were significantly higher than other two groups. LIPUS-BM-MSC exosomes significantly inhibited fat infiltration, promoted cell proliferation and formation of fibrocartilage and proteoglycan, further promoted extracellular matrix deposition and repaired the tendon–bone interface.	LIPUS-BM-MSC exosomes up-regulated the expression of chondrogenic genes such as COL II, SOX-9, and aggregate, and down-regulated the expression of adipogenic genes such as Adipo, Retn, and Pparg by miR-140.
Xu et al. [81]	2022	In the IMEI group, the width of the interface was smaller, the inflammatory reaction was lighter, the fibrocartilage formation was obvious, and the collagen fibers were more orderly arranged. The IMEI group had the smallest mean bone tunnel area and the highest BV/TV. Exosomes significantly increased the ultimate failure load and stiffness, and promoted the M2 polarization of macrophages.	NR
Zhang et al. [82]	2022	The Hypo-Exo group had the smallest mean bone tunnel area, the largest BV/TV, more fibrocartilage formation, and the highest histological score. Exosomes increased the ultimate failure load and stiffness of tendon, and promoted the formation of CD31^+^/Emcn^+^ blood vessels at the tendon–bone interface. Hypo-exosomes were the most effective.	Rat BM-MSC exosomes can promote the proliferation, migration and tube formation of HUVECs, and Hypo-exosomes were the most effective.

BM-MSC, bone morrow mesenchymal stem cells; COL I: Collagen type I; COL II, Collagen type II; VEGF, Vascular endothelial growth factor; HUVECs, human umbilical vein endothelial cells; SCX, Scleraxis; IMEI, Injection of exosome from Infrapatellar fat pad mesenchymal stem cell; BV/TV, bone volume/total volume; Hypo-exosomes, exosomes secreted by hypoxia-stimulated bone-marrow mesenchymal stem cells; TGF-β: Transforming growth factor-β; IGF, insulin-like growth factors; α-SMA: Alpha-smooth muscle actin; OCN, osteocalcin; COL III, Collagen type III; IONP-exosomes, Exosomes derived from magnetically actuated bone morrow mesenchymal stem cells; Smad, Mothers Against Decapentaplegic Homolog; BMD, bone mineral density; TMD, tissue mineral density; BMP, Bone morphogenetic protein; EXO, exosomes; RUNX, Runt-related Transcription Factor; SOX-9, sex determining region Y box 9; TIMP, Tissue inhibitor of the metalloproteinase; TNMD, Tenomodulin; TNC, Tenascin C; DCN, Decorin; EGR, early growth response; Spp1, Secreted Phosphoprotein 1; PPARG, peroxisome proliferator activated receptor gamma; LIPUS-BM-MSC, Bone Marrow Mesenchymal Stem Cell Preconditioned by Low-Intensity Pulsed Ultrasound Stimulation; Adipo, adiponectin; Retn, resistin; Pparg, peroxisome proliferator-activated receptor-g.

## Data Availability

Not applicable.

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
