# Peer review of "Therapeutic Potential of Exosomes in Tendon and Tendon–Bone Healing: A Systematic Review of Preclinical Studies"

_jfb, 2023, doi:10.3390/jfb14060299_

Round 1
Reviewer 1 Report
The authors impressed the reviewer with their very well written Systematic Review entitled "Therapeutic Potential of Exosomes in Tendon and Tendon-bone Healing: a Systematic Review of Preclinical Studies". This can also be published after a few minor corrections. It has been extensively researched on a very important topic for the future tissue engineering approach of tendon regeneration.
Please correct:
Line 60: why there some hyphens written after "zone --"? do you need them? if not the reviewer would suggest to delete it.
Figure 1: please have a look at the point 8. the distance between number and the text is not the same as the other bullet points. and also in the box below with the bullet point 1. the number and the text have a larger distance than the other point.
Line 222: the reviewer would suggest to write Fifteen (large F) due to starting the sentence.
Line 228: the reviewer would suggest to write the animals (rats, mice, rabbits, dogs and sheep) small and not upper-case letter.
in general for Table 2 and 4: the reviewer would be interested which systematic system you use for the Table 2 and 4? The Reviewer could not see an alphabetical, yearly or orther order.
Line 294: the reviewer would suggest to write adipose MSCs
Line 294: IFN is mentioned the first time in Line 294, the reviewer would suggest to write out interferon and put the abbreviation in parentheses.
Line 484: the reviewer would suggest to write "preconditioned"
Line 544: The reviewer would suggest to make a space between "passage 12" (you did this also between "passage 5")
The reviewer would like to thank the authors for this very interesting and extremely well-written manuscript and wishes them every success in publishing it
Best regards
Author Response
Response to Reviewer 1 Comments
Point 1: Line 60: why there some hyphens written after "zone --"? do you need them? if not the reviewer would suggest to delete it.
Response 1: We thank the reviewer for raising this question. We have deleted the hyphens in line 60.
Point 2: Figure 1: please have a look at the point 8. the distance between number and the text is not the same as the other bullet points. and also in the box below with the bullet point 1. the number and the text have a larger distance than the other point.
Response 2: We thank the reviewer for raising this question. We have already adjusted the distance between the numbers and the text.
Point 3: Line 222: the reviewer would suggest to write Fifteen (large F) due to starting the sentence.
Response 3: We thank the reviewer for raising this question. We have change “fifteen” into “Fifteen”.
Point 4: Line 228: the reviewer would suggest to write the animals (rats, mice, rabbits, dogs and sheep) small and not upper-case letter.
Response 4: We thank the reviewer for raising this question. We have changed the corresponding uppercase letters in line 228 to lowercase letters.
Point 5: In general for Table 2 and 4: the reviewer would be interested which systematic system you use for the Table 2 and 4? The Reviewer could not see an alphabetical, yearly or orther order.
Response 5: We thank the reviewer for raising this question. We are very sorry, and we have rearranged the table in alphabetical order.
Point 6: Line 294: the reviewer would suggest to write adipose MSCs.
Response 6: We thank the reviewer for raising this question. We have changed “Adipose MSCs” into “adipose MSCs” in line 294.
Point 7: Line 294: IFN is mentioned the first time in Line 294, the reviewer would suggest to write out interferon and put the abbreviation in parentheses.
Response 7: We thank the reviewer for raising this question. We have added 'interferon' in line 294 and placed the abbreviation in parentheses.
Point 8: Line 484: the reviewer would suggest to write "preconditioned".
Response 8: We thank the reviewer for raising this question. We have changed “Preconditioned” into “preconditioned” in line 484.
Point 9: Line 544: The reviewer would suggest to make a space between "passage 12" (you did this also between "passage 5")
Response 9: We thank the reviewer for raising this question. We have made a space between “passage 12” in line 544.

Reviewer 2 Report
Specific Comments
(1) Abstract: Results: Numerical data on findings should be integrated into the abstract.
(2) Introduction (Line 45): The wording “….oral anti-inflammatory medications, surgery and so on[7-10].” Is vague. I would recommend to be more specific on treatment options for acute and chronic tendon injuries. A table or figure can add important information on current accepted and validated tendon treatment options.
(3) Introduction (Lines 45-47): I would recommend excluding surgical repair options as this is not under the scope of the review.
(4) Introduction (Lines 68-70): I would recommend expanding the cellular regenerative treatment options. Currently various forms of cellular therapies are used to treat tendon injuries including PRP, SVF and Stem Cells. All have exosomes. A group of tables highlighting clinical studies for such cellular therapies could be very important.
(5) Results: How about clinical outcomes? There are commercially available exosome isolation technologies. There are clinical studies registered to clinicaltrials.gov. I would suggest authors to re-review clinical studies and add to this systemic review.
Author Response
Response to Reviewer 2 Comments
Point 1: Abstract: Results: Numerical data on findings should be integrated into the abstract.
Response 1: We thank the reviewer for raising this question. We have supplemented numerical data (quantitative data). All animal types and corresponding animal numbers used in the study are reported. It should be noted that in section 2.4 and discussion part of the manuscript, we pointed out that due to the insufficient quantitative data provided by the included studies and the heterogeneity caused by inconsistent format of the reports, quantitative data analysis is limited. Therefore, the studies were largely evaluated qualitatively.
Point 2: Introduction (Line 45): The wording “….oral anti-inflammatory medications, surgery and so on[7-10].” Is vague. I would recommend to be more specific on treatment options for acute and chronic tendon injuries. A table or figure can add important information on current accepted and validated tendon treatment options.
Response 2: We thank the reviewer for raising this question. We have added a table to specify the current treatment options for acute and chronic tendon injuries.
Point 3: Introduction (Lines 45-47): I would recommend excluding surgical repair options as this is not under the scope of the review.
Response 3: We thank the reviewer for raising this question. We have completely replaced the text content of lines 45 to 47, excluding surgical options.
Point 4: Introduction (Lines 68-70): I would recommend expanding the cellular regenerative treatment options. Currently various forms of cellular therapies are used to treat tendon injuries including PRP, SVF and Stem Cells. All have exosomes. A group of tables highlighting clinical studies for such cellular therapies could be very important.
Response 4: We thank the reviewer for raising this question. We have added a table to summarize the existing cellular regenerative treatment options for tendon injuries.
Point 5: Results: How about clinical outcomes? There are commercially available exosome isolation technologies. There are clinical studies registered to clinicaltrials.gov. I would suggest authors to re-review clinical studies and add to this systemic review.
Response 5: We thank the reviewer for raising this question. We have reviewed clinical studies about exosomes on clinicaltrials.gov and cochranelibrary. However, there is no clinical trial of exosomes promoting tendon regeneration, nor is there any clinical study registration information of exosomes promoting tendon regeneration.

Reviewer 3 Report
Several comments given for the submitted systematic review as follows:
1. Please make sure the authors have been followed PRISMA 2020 precisely.
2. Literature resource, encouraged only to use three main databases, scopus, web of science, and pubmed.
3. Quantitative results must be included in the abstract section.
4. Please add the abstract's "take-home" message, the current form was insufficient.
5. Reorder keywords based on alphabetical order.
6. It is suggested to not use abbreviations in the keywords.
7. The Reviewer does not see the novel in the present article. My examination revealed that several similar previous publications appear to appropriately address the issues you have brought up in the current submission. Please emphasize it more advance in the introduction section if there are any more truly something really new.
8. Please explain the urgency of biomaterials selection in medical application. For this purpose, please refer the relevant reference as follows, doi: 10.3390/biomedicines11030951, 10.3390/ma14247554, and 10.3390/su142013413
9. Encouraging to make the objective of the present work more clearly.
10. Additional figures in the introduction would improve the quality of the present article. Please provide it.
11. Rather than relying just on the predominant text as it already exists, the authors could incorporate more illustrations as figures in the materials and methods section that illustrate the workflow of the current study.
12. Please, that major improvement is needed in the discussion section of the present manuscript, where the present form is not enough.
Author Response
Response to Reviewer 3 Comments
Point 1: Please make sure the authors have been followed PRISMA 2020 precisely.
Response 1: We thank the reviewer for raising this question. We have strictly followed PRISMA 2020. It should be noted that in section 2.4 and discussion part of the manuscript, we pointed out that due to the insufficient quantitative data provided by the included studies and the heterogeneity caused by inconsistent format of the reports, quantitative data analysis is limited. Therefore, the studies were largely evaluated qualitatively. In the PRISMA2020 checklist, we also marked the quantitative analysis section as “Not applicable”.
Point 2: Literature resource, encouraged only to use three main databases, scopus, web of science, and pubmed.
Response 2: We thank the reviewer for raising this question. Indeed, the included studies mainly come from pubmed, web of science and scopus.
Point 3: Quantitative results must be included in the abstract section.
Response 3: We thank the reviewer for raising this question. We have supplemented quantitative data. All animal types and corresponding animal numbers used in the study are reported. It should be noted that in section 2.4 and discussion part of the manuscript, we pointed out that due to the insufficient quantitative data provided by the included studies and the heterogeneity caused by inconsistent format of the reports, quantitative data analysis is limited. Therefore, the studies were largely evaluated qualitatively.
Point 4: Please add the abstract's "take-home" message, the current form was insufficient.
Response 4: We thank the reviewer for raising this question. We have added the following "take home" messages in abstract: 1. We have outlined the mechanisms by which exosomes promote tendon healing and tendon bone healing. 2. We pointed out the limitations of current studies and the future research directions.
Point 5: Reorder keywords based on alphabetical order.
Response 5: We thank the reviewer for raising this question. We have reordered keywords based on alphabetical order.
Point 6: It is suggested to not use abbreviations in the keywords.
Response 6: We thank the reviewer for raising this question. We did not use any abbreviation in the keywords.
Point 7: The Reviewer does not see the novel in the present article. My examination revealed that several similar previous publications appear to appropriately address the issues you have brought up in the current submission. Please emphasize it more advance in the introduction section if there are any more truly something really new.
Response 7: We thank the reviewer for raising this question. Compared to previous publications, we have included more literature and this systematic review is more comprehensive. Currently, the latest publication on this topic was from 2021, while our review added more up-to-date evidence to the synthesis of literature. Besides, we systematically reviewed the signaling pathways involved in exosomes promoting tendon healing and tendon bone healing, as well as the expression changes of related genes, and further elucidated their mechanisms. In addition, we also analyzed the heterogeneity in different studies with possible explanations, providing advice for future research.
Point 8: Please explain the urgency of biomaterials selection in medical application. For this purpose, please refer the relevant reference as follows, doi: 10.3390/biomedicines11030951, 10.3390/ma14247554, and 10.3390/su142013413
Response 8: We thank the reviewer for raising this question. We explained the urgency of biomaterials in medical applications in the fifth paragraph of the Introduction and cited the three references the reviewer has mentioned.
Point 9: Encouraging to make the objective of the present work more clearly.
Response 9: We thank the reviewer for raising this question. We have added the following "take home" messages in abstract to make our objective of the work more clearly: 1. We have outlined the mechanisms by which exosomes promote tendon healing and tendon bone healing. 2. We pointed out the limitations of current studies and future research directions. In addition, we have revised the introduction and discussion sections to further clarify our research objectives (see the last paragraph of the introduction). We also summarized the therapeutic effect of exosomes on tendon injury at the beginning of the discussion section.
Point 10: Additional figures in the introduction would improve the quality of the present article. Please provide it.
Response 10: We thank the reviewer for raising this question. We have added two tables in the introduction to provide an overview of current treatment options for acute and chronic tendon injuries, as well as cellular regenerative treatment options for tendon injuries.
Point 11: Rather than relying just on the predominant text as it already exists, the authors could incorporate more illustrations as figures in the materials and methods section that illustrate the workflow of the current study.
Response 11: We thank the reviewer for raising this question. We are very sorry, as the time for major revisions is only 10 days, we are unable to provide illustrations in such a short time. In future studies, we will consider making illustrations as soon as possible.
Point 12: Please, that major improvement is needed in the discussion section of the present manuscript, where the present form is not enough
Response 12: We thank the reviewer for raising this question. We have tried our best to make major improvement on the discussion section. We adjusted the structure of the discussion and added more information.

Round 2
Reviewer 3 Report
Some major comments still needs to addressed.
1. Line 41-45, please develop this explanation, just not focus in number of patient.
2. Line 63, rotor cuff tear needs to explained the concept.
3. Line 68 for table 1, the difference is not clear.
4. Line 181, I am not sure the presented figure have been followed PRISMA 2020. Please recheck again.
5. The limitation of the present submission needs to be added at the end of the discussion section before entering the conclusion section.
6. Please discuss the further research in the conclusion section.
7. The reference should be enriched with literature from the last five years.
8. Please change the reference number [29] with a more relevant one.
9. Please reduce the literature used as a reference that is authored by the present author in order to reduce the number of self-citation.
10. English needs to be proofread due to grammatical errors and English style.
11. The graphical abstract should be provided in the system after modification of peer review.
12. The authors encouraged to incorporated bone regenerative in scaffold used effort.
13. Potential further study adopting computational simulation/in silico needs to discusses. It would become preclinical study that provide preliminary results. Also, it bring several advantages compared to clinical study such as lower cost and faster results. The results from computational simulation/in silico also would support the clinical results.
Author Response
Response to Reviewer 3 Comments
Point 1: Line 41-45, please develop this explanation, just not focus in number of patient.
Response 1: We thank the reviewer for raising this question. We have develop the explanation by describing the structure of tendons and emphasizing the regeneration of injured tendons.
The revision to this issue is now located on the first paragraph of Introduction.
Point 2: Line 63, rotor cuff tear needs to explained the concept.
Response 2: We thank the reviewer for raising this question. We have provided further explanation for the conception of rotator cuff tear.
The revision to this issue is now located on line 82~84 (word)
Point 3: Line 68 for table 1, the difference is not clear.
Response 3: We thank the reviewer for raising this question. We have changed the title of table 1 and revised the content of table 1 to make the difference clear.
The revision to this issue is now located on line 89 and Table 1 (word)
Point 4: Line 181, I am not sure the presented figure have been followed PRISMA 2020. Please recheck again.
Response 4: We thank the reviewer for raising this question. We have rechecked the figure, and we make sure that it followed PRISMA 2020.
Point 5: The limitation of the present submission needs to be added at the end of the discussion section before entering the conclusion section.
Response 5: We thank the reviewer for raising this question. We have already discussed the limitations of this system review at the end of the discussion section (4.8)
The revision to this issue is now located on line 945~964 (word)
Point 6: Please discuss the further research in the conclusion section.
Response 6: We thank the reviewer for raising this question. We have discuss the future research on this topic in the conclusion section.
The revision to this issue is now located on line 970~977 (word)
Point 7: The reference should be enriched with literature from the last five years.
Response 7: We thank the reviewer for raising this question. We have added some new literature from the last five years and We have replaced some old literature with literature from the last five years.
Point 8: Please change the reference number [29] with a more relevant one.
Response 8: We thank the reviewer for raising this question. We have changed the reference with a more relevant one.
Point 9: Please reduce the literature used as a reference that is authored by the present author in order to reduce the number of self-citation.
Response 9: We thank the reviewer for raising this question. We did not cite any articles written by the authors of this article.
Point 10: English needs to be proofread due to grammatical errors and English style.
Response 10: We thank the reviewer for raising this question. We have rechecked the whole passage, and tried our best to correct the grammatical errors and improved the English style.
Point 11: The graphical abstract should be provided in the system after modification of peer review.
Response 11: We thank the reviewer for raising this question. We will provide the graphical abstract in the system.
Point 12: The authors encouraged to incorporated bone regenerative in scaffold used effort.
Response 12: We thank the reviewer for raising this question. This systematic review mainly focuses on tendon healing and tendon bone healing. Therefore, we did not include relevant content on bone regeneration. Indeed, using scaffolds to promote bone regeneration has great prospects, and we will focus on this field in future research.
Point 13: Potential further study adopting computational simulation/in silico needs to discusses. It would become preclinical study that provide preliminary results. Also, it bring several advantages compared to clinical study such as lower cost and faster results. The results from computational simulation/in silico also would support the clinical results.
Response 13: We thank the reviewer for raising this question. We have discussed the potential further study adopting computational simulation/in silico in the discussion section.
The revision to this issue is now located on line 938~944 (word)